

# 1 Seasonal Net Ecosystem Metabolism of the Near-Shore Reef System in La

# 2 Parguera, Puerto Rico

Melissa Meléndez[1]
Joseph Salisbury[1]
Dwight Gledhill[2]
Chris Langdon[3]
Julio M. Morell[4,5]
Derek Manzello[6]
Sylvia Musielewicz[7,8]
Adrienne Sutton[7]
Affiliation and address
[1]Department of Earth Sciences and Ocean Processes Analysis Laboratory, University of New Hampshire, Durham,
New Hampshire, 03824, USA
[2]National Oceanic and Atmospheric Administration (NOAA), Ocean Acidification Program, Silver Spring,
Maryland, 20910, USA
[3]Rosenstiel School of Marine and Atmospheric Science, University of Miami, Miami, FL, USA
[4]Department of Marine Sciences, University of Puerto Rico, Mayagüez, Puerto Rico, 00680, USA
[5]Caribbean Coastal Ocean Observing System, NOAA, Magueyes Island, Lajas, Puerto Rico, 00667, USA
[6]Atlantic Oceanographic and Meteorological Laboratory, NOAA, Miami, Florida, USA
[7]Pacific Marine Environmental Laboratory, NOAA, Seattle, Washington, 98115, USA
[8]University of Washington Joint Institute for the Study of the Atmosphere and Ocean, Seattle, WA 98195, USA
Correspondence to: Melissa Meléndez (mm19@wildcats.unh.edu)
**Keywords:** Puerto Rico, Coral Reef, Net Ecosystem Calcification, Net Ecosystem Production,
Dissolution, Carbonate Chemistry



## Abstract

Changes in ocean chemistry as a direct response to rising atmospheric carbon dioxide ($CO_2$) concentrations is causing a reduction of pH in the surface ocean. While the dynamics and trends in carbonate chemistry are reasonably constrained for open ocean waters, the ways in which ocean acidification (OA) manifests within the shallow near-shore waters, where coral reefs reside, is less understood. Constraining near-reef variability in carbonate chemistry and net ecosystem metabolic processes across diel, seasonal, and annual scales is important in evaluating potential biogeochemical thresholds of OA that could result in ecological community changes. The OA Test-Bed at La Parguera Marine Reserve in Puerto Rico provides long-term carbonate chemistry observations at high-temporal resolution within a Caribbean near-shore coral reef ecosystem. A 1-D model was developed using the carbon mass balance approach to yield information about net ecosystem production and calcification processes occurring in the water column adjacent to the reef. We present results of nine years of sustained monitoring at the Enrique mid-shelf forereef, which provides for the characterization of temporal dynamics in carbonate chemistry and net ecosystem metabolic processes encompassing near-shore and upstream locations. Results indicate that net heterotrophy and net dissolution dominate over most of the year, while net autotrophic conditions coupled with calcification dominated from only January to mid-April. The average carbonate dissolution rate observed during summer is estimated at -2.19 g $CaCO_3$ m$^{-2}$ day$^{-1}$ and net community dissolution persists 76 % of the seasonal year despite the water column remaining super-saturated with respect to aragonite. This corresponds to -0.62 kg $CaCO_3$ m$^{-2}$ year$^{-1}$, classifying the Enrique fore-reef and off-reef areas in a net dissolutional state. The combination of thermodynamically-driven depressed aragonite saturation state and high rates of respiration during the summer cause conditions that jeopardize the most soluble carbonate minerals and the free energy in the system for calcification. These data suggest that the reef area and associated ecosystems upstream of the sampling location are experiencing a net loss of $CaCO_3$, possibly compromising coral ecosystem health and reef accretion processes necessary for maintenance as sea level increases. Resiliency from other climate-scale stressors including rising sea surface temperatures and coral bleaching is likely to be compromised in a system exhibiting net carbonate loss.





## 1 Introduction


Oceans are the largest natural sink for excess atmospheric carbon dioxide ($CO_{2,air}$), currently absorbing
approximately one-third (~ 28 %) of all anthropogenic $CO_{2,air}$ each year (Sabine et al., 2004; Le Quéré et
al., 2016). Under $CO_2$ "baseline scenarios" of the Intergovernmental Panel on Climate Change (IPCC)
Representative Concentration Pathways (RCP6.0 and RCP8.5), dissolved $CO_2$ in the surface ocean
($CO_{2,sw}$) will likely double over its pre-industrial concentration (~280 µatm) by the middle of this century,
producing significant changes in the ocean carbonate chemistry (IPCC, 2014).

As $CO_{2,air}$ is absorbed by the surface ocean it reacts to form carbonic acid ($H_2CO_3$) which rapidly
dissociates to hydrogen ion ($H^+$) and bicarbonate ion ($HCO_3^-$), thereby lowering pH and hence the term
"acidification". The resulting change in pH is buffered in part by the reaction of $H^+$ with carbonate ions
($CO_3^{2-}$) forming bicarbonate. The low pH favors bicarbonate over carbonate thereby driving a net
depletion of carbonate ions, as given by:

$$CO_{2,air} \leftrightarrow CO_{2,sw} + H_2O \leftrightarrow \underbrace{H_2CO_3}_{Carbonic\ Acid} \leftrightarrow H^+ + \underbrace{HCO_3^-}_{Bicarbonate} \leftrightarrow 2H^+ + \underbrace{CO_3^{2-}}_{Carbonate} \qquad (1)$$


This chemical process affects the degree to which seawater is saturated with respect to calcium carbonate
($CaCO_3$) minerals:

$$Ca^{2+} + CO_3^{2-} \leftrightarrow CaCO_{3,solid} \qquad (2)$$


where, $Ca^{2+}$ is the calcium ion. The solubility of $CaCO_3$ minerals in seawater is described by Mucci
(1983) and is defined according to the stoichiometric solubility product constant ($K^*_{SP}$) as a function of
temperature, pressure, and solution composition. The degree of carbonate saturation with respect to the
$CaCO_3$ mineral phase of interest (e.g., calcite, aragonite, Mg-calcite) is indicated by the saturation state
($\Omega$). Aragonite and calcite represent the most important calcium carbonate polymorphs, but Mg-calcite
can contain variable amounts of magnesium ion ($Mg^{2+}$) ranging from 10 to >30% (Morse et al., 2007).





The $\Omega$ is defined as the ratio of concentration product to $K^*_{SP}$:

$$\Omega_x = \frac{[Ca^{2+}]\,[Mg^{2+}][CO_3^{2-}]}{K^*_{sp}}$$   (3)

where, [ ] indicates total concentrations. Values of $\Omega > 1$ are typical for tropical surface ocean waters and
indicate that precipitation of the mineral phase is favored; if $\Omega < 1$ dissolution is favored; and if $\Omega = 1$ the
mineral is in equilibrium with seawater and no net dissolution nor precipitation is expected. The energy
available in the system to drive mineral precipitation can be related to $\Omega$ according to:

$$\Delta G = RT\ln\Omega_x = RT\ln\frac{IAP}{k_x}$$   (4)

where, $\Delta G$ is the Gibbs free energy (cal mol$^{-1}$ CaCO$_3$), $R$ is the ideal gas constant (J mol$^{-1}$ K$^{-1}$), $T$ is
absolute temperature (K), $IAP$ is the ion activity product, and $k$ is the thermodynamic solubility product.
The subscript $x$ denotes the mineral phase of interest. While regions of the Southern Ocean surface waters
could experience under-saturation with respect to aragonite (i.e., $\Omega_{arag} < 1$), should $CO_{2,air}$ levels exceed
500 µatm, tropical surface oceanic waters are projected to remain supersaturated (i.e., $\Omega_{arag} > 1$) for several
centuries (Cao and Caldeira, 2008). Despite this continued super-saturation, concerns remain regarding
the energy required by calcifers to sustain calcification and the free energy available in the system to drive
mineral precipitation with continued OA (Cohen and Holcomb, 2009).

Multiple laboratory studies have reported reduced rates of calcification for many species of reef-building
coral as a function of changes in carbonate chemistry and $\Omega_{arag}$ (e.g., Borowitzka 1981; Gao et al., 1993;
Gattuso et al., 1998; Langdon et al., 2000, 2003; Leclercq et al., 2000, 2002; Marubini et al., 2001;
Marshall and Clode, 2002; Ohde and Hossain, 2004; Langdon and Atkinson, 2005; Anthony et al., 2008;
Chan and Connolly, 2013). The rates of calcification (carbonate production) respond negatively, while
biologically-mediated and metabolic dissolution respond positively to declining $\Omega_{arag}$, representing not a
"switch" (i.e., threshold effect) response, but rather a "dimmer" effect (Andersson and Gledhill, 2013).



Experimental studies have also revealed that $\Omega_{arag}$ has an indirect effect on coral recruitment by disrupting
larval-algal settlement interactions (Doropoulos et al., 2012) and direct effects on recruitment (Albright
et al., 2010) which could prove far more disruptive under the combination of warming and elevated
seawater $CO_2$ than under either stressor singly (Albright and Mason, 2013).

These changes will have implications for shallow water $CaCO_3$ mineral kinetics (Morse and Mackenzie,
1990), sediment shelf compositions, and potentially, marine ecosystems dependent upon skeletal and shell
structures comprised of $CaCO_3$ minerals (Kleypas et al., 1999; Riebesell et al., 2000). The net loss of
$CaCO_3$ materials may induce changes in coastlines that are directly created by $CaCO_3$ (i.e., calcareous
sand beaches), or those that are shielded from hydrodynamic disturbance due to the baffling effect of
offshore $CaCO_3$ ecosystems like coral reefs (Lowe et al., 2009; Andersson and Gledhill, 2013; Comeau
et al., 2016). Ecosystems for which the formation of $CaCO_3$ substrates serve an important function, such
as shallow water coral reefs, are of concern under continued OA as they provide a vital habitat for
numerous other organisms and provide economic, cultural, and social benefits (e.g., Lugo-Fernández et
al., 1998; Gratwicke and Speight, 2005).

Tropical surface ocean waters in the Caribbean show that calcification rates may have dropped by ~15 %
with respect to their pre-industrial values (Friedrich et al., 2012), and $\Omega_{arag}$ is currently decreasing at a
rate of about 3 % per decade (Gledhill et al., 2008; Bates et al., 2012). Perry et al. (2018) found that many
shallow coral reefs across the Caribbean are very close to carbonate production and accretion thresholds
due to the decline in live coral cover and sea level rise. However, the accretion rates and thresholds
presented in Perry et al. (2018) may be lower since reef physical erosion and chemical dissolution were
not included in the projections. Tropical near-shore coral reefs could have transitioned from net
precipitation to net dissolution (e.g., Muehllehner et al., 2016), but attributing changes in net ecosystem
metabolic rates to variations in carbonate chemistry remains challenging in coral reef ecosystems due to
the multiple biogeochemical processes affecting coastal waters (Duarte et al., 2013).



Forecasting the effects of continued OA on coral reefs has relied heavily on laboratory and small-scale
mesocosm manipulative experiments. The results of these experimental manipulations are often applied
to projections of surface ocean carbonate chemistry derived from coupled climate/carbon-cycle models
(e.g., Cao and Calderia, 2008). However, most coastal systems exhibit considerably greater complexity
than those reproducible in laboratories and expressed in these global models. Seasonal and diurnal
variations in seawater carbonate chemistry and $\Omega_{arag}$ within the reef zone can be several times higher or
lower than expected for oceanic waters due to net ecosystem metabolic processes alone (Gattuso et al.,
1993; Bates et al., 2010; Turk et al., 2015; Page et al., 2017). The net ecosystem production (NEP, the
sum of gross production and respiration processes from all the autotrophic and heterotrophic components
of the system) alters the dissolved inorganic carbon (DIC) and the [$H^+$] concentration and thus can alter
the relative partitioning of the carbonate species (Eq.1) and $\Omega_{arag}$. Net ecosystem calcification (NEC, the
sum of calcium carbonate production and dissolution processes from all biotic and abiotic components of
the system) alters the seawater total alkalinity (TA) and DIC in a ratio of 1:2. The combination of NEP
and NEC (under constant temperature, salinity, and pressure), change the seawater DIC:TA ratio, which
dictates properties such as pH, $CO_2$ partial pressure ($p\text{CO}_{2,sw}$), and $\Omega$ (Zeebe and Wolf-Gladrow, 2001).

NEC and NEP measurements describe the net accumulation or consumption of inorganic and organic
carbon, respectively (e.g., Gattuso et al., 1998). NEC can also be described in terms of the $CaCO_3$ gain
and loss due to calcification and chemical dissolution processes. This is necessary to consider in order to
quantify the total net accretion rates of the reef substrate (e.g., Eyre et al., 2014). Any decline in NEC
relative to NEP could compromise many reef systems, since rates of accretion on healthy, undisturbed
reefs only slightly outpace rates of reef loss due to physical and biological erosion (see Glynn and
Manzello, 2015). Gradual changes in NEP and NEC can bring the system closer to a 'tipping point' where
even small perturbations can cause a shift on the autotrophic-calcifying balance that characterized coral
reef ecosystems, to a system dominated by heterotrophic and dissolving conditions (Silbiger et al., 2014;
Yeakel et al., 2015). However, discerning whether these net ecosystem metabolic processes are changing
due to natural ecological variability (e.g., changes in benthic communities) or because of anthropogenic



activities (e.g., eutrophication, OA) is non-trivial and demands sustained robust monitoring at a high
temporal resolution, requiring *in situ* autonomous measurements and *in situ* water sampling.

Despite the importance of monitoring NEC and NEP using standard best-practices for sustained
monitoring of these rates remains a major challenge. NEC and NEP measurements typically adopt various
Lagrangian or Eulerian techniques requiring high-temporal resolution of *in situ* bottle water samples (e.g.,
Gattuso et al., 1996; Bates, 2002; Silverman et al., 2007). Recently, a few *in situ* autonomous methods
have been proposed which offer great promise, but currently remain cost prohibitive for most networks,
and have a limited duration of a few weeks at most due to biofouling and sensor drift (e.g., Yates et al.,
1999; McGillis et al., 2011; Falter et al., 2008; Takeshita et al., 2016). Nevertheless, techniques and
methods that incorporate existing observations of carbonate chemistry presently allow the development
of net ecosystem metabolic proxies that could be employed to monitor these biogeochemical fluxes.

The time series presented here benefits from nine years (from 2009 to 2017) of continuous observations
of the carbonate chemistry at the forereef of Enrique coral reef in Puerto Rico. The combined
measurements obtained from the moored autonomous $p$CO$_2$ (Ma$p$CO$_2$) buoy technology with a dedicated
discrete sampling campaign have provided high-quality observations allowing reasonable constraint of
the carbonate system and of the biological processes affecting the water mass as it flows in over the reef
platform from the open ocean end member. Moreover, the time series enables us to derive multi-year
estimates of NEP and NEC using site-specific empirical relationships derived from both *in situ*
autonomous and discrete bottle data.

In this manuscript, we described the seasonal variability and primary controls of the carbonate system
variability and how such dynamics may change as a consequence of OA in the La Parguera near-shore
reef system. In order to accomplish this, we tailored our approach to de-couple the biological processes
on NEP and NEC, and derived seasonal net ecosystem metabolic rates for the reef system mixed layer
using a one-dimensional (1-D) mass balance model developed with the autonomous Ma$p$CO$_2$ buoy
observations. These metabolic observations serve to provide a more comprehensive interpretation of how



near-shore carbonate chemistry is governed by carbon cycling and to enhance our understanding of the
potential long-term impacts of OA on carbonate and energy budgets.

## 2 Methods

### 2.1 La Parguera OA Time Series

The La Parguera OA time series (Fig.1) is an on-going project advanced by the NOAA Coral Reef
Conservation Program (CRCP) that was established in 2009. The test-bed has since been adopted as a
long-term sustained monitoring station providing physical, chemical, and ecological data under the
National Coral Reef Monitoring Program (NCRMP, www.coris.noaa.gov/monitoring/) jointly sponsored
by NOAA CRCP and the NOAA Ocean Acidification Program (OAP). Details about the efforts
supporting this time series station are described in the supplemental material, S1.

### 2.2 Study sites

The Marine Reserve of La Parguera is located on the southwest coast of Puerto Rico (Fig.2-a). The well-
developed reef system consists of different habitat types dominated by seagrasses, macroalgal beds,
unconsolidated carbonate sediments, and mangroves. The reserve exists in an area with relatively low
coastal development, an absence of local rivers, and the lowest rainfall rate in Puerto Rico (Pittman et al.,
2010). Local environmental stressors include coastal runoff and fishing (Garcia-Sais et al., 2008).

The insular shelf extends 10-11 km from the coast and is divided in three different regions (inner, middle,
and outer shelf reefs) according to its general morphological and depositional characteristics (details in
Morelock et al., 1977). The Ma$p$CO$_2$ buoy is located at the west-end of the Enrique middle-shelf reef at
2.5 km from the coast and over the forereef where the water depth is about 3 m (Fig.2-c).

Cayo Enrique is an emergent reef adjacent to a mangrove system. The back reef and reef flat areas consist
of seagrass beds dominated by *Thalassia testudinum* and a few coral patch reefs. The forereef and the
upper zone benthic reef communities at Enrique were formerly dominated by the branching coral
*Acropora palmata*. Significant live coral cover was reduced by >50 % from 2003 to 2007 (Appeldoorn,




2009). Losses in coral cover are attributed to coral bleaching and disease, and secondarily from
hurricanes, and corallivorous mollusks (Morelock et al., 2001; Ballantine et al., 2008). The die-off of the
long-spined sea urchin, *Diadema antillarum* limited coral recovery and prevented recruitment to bare
substrates due to the increased algae cover (Ballantine et al., 2008). Currently, most of the area formerly
covered by *A. palmata* is dead and replaced by *Millepora* spp., *Palythoa caribaeorum* (zoanthids), and
soft corals such as *Gorgonia* spp. and *Pseudopterogorgia* spp. McGillis et al. (2011) and Manzello et al.
(2017) noted approximately 10 and 11 % live coral cover area during belt transect surveys at the Ma$p$CO$_2$
area in March 2009 and August 2015, respectively. The six major reef building corals reported by
McGillis et al. (2011) were: *Siderastrea siderea*, *Porites astreoides*, *Pseudodiploria strigosa*, *Siderastrea*
*radians*, *Pseudodiploria clivosa*, and *Porites porites* (ranked in order of areal cover). The dominant
benthic communities reported by Manzello et al. (2017) were: sand (30%), soft coral (25%), turf algae
(17%), sponges (10%), and coral rubble (5%). A survey conducted in 2011 by Moyer et al. (2012), showed
that the non-calcifying algal cover was ~ 25 %, with little seasonal variation. However, Pittman et al.
(2010) reported a peak of turf algae and macroalgal cover in the summer. Calcifying algae (crustose
coralline algae) represent about 1 % of the benthic cover with no significant seasonal variation (Pittman
et al., 2010; Moyer et al., 2012).

The sediments at the forereef of Enrique consist predominantly of CaCO$_3$, with lesser amounts of
terrigenous (<10 %) and organic material (Ryan et al., 2008; Hernández et al., 2009). According to
Morelock et al. (1977) the major CaCO$_3$ contributors are fragments of calcareous green algae (*Halimeda*
spp.), coral fragments (40 to 80 %), and coralline-algae (10 to 20 %). It is possible that the relative
proportions of the sediment composition and re-distribution have changed over the last 40 years during
high-wave energy events (e.g., hurricanes and storms). However, Hernández et al. (2009) found that
CaCO$_3$ minerals are still the predominant sediment type at the site, dominated by aragonite and Mg-calcite
with 13-14 mol % MgCO$_3$, and with lesser amounts of ~3 mol % MgCO$_3$. These are produced *in situ* by
bioerosion and mechanical processes (Hernández et al., 2009).



NEP rates were estimated at Enrique forereef during March of 2009 using the boundary layer $O_2$ gradient
and enclosure methods (McGillis et al., 2009; McGillis et al., 2011). The average daily NEP rate were
43.1 and 60.3 mmol C m$^{-2}$ d$^{-1}$ (converted from $O_2$ to C using the Redfield molar ratio of 138:106) for
both methods respectively (McGillis et al., 2011). Results from both methods suggested that during the
time of these experiments organic carbon respiration contributed $CO_2$ to reef waters. During the same
study the ocean current speeds were measured using a set of ADCPs. The average current speed ranged
between 2 to 10 cm s$^{-1}$ towards the west.

Reef areas are flushed by offshore water and typically have water residence times of days to weeks, while
longer residence times (months) are found closer to shore (Venti et al., 2012). The residence times from
the offshore station and Enrique reef (Fig.2) were measured in 2011 using the inputs and outputs of
Beryllium-7 ($^7$Be) according to Venti et al. (2012, 2014). The mean residence times calculated for January
and May were 9 and 11 ± 2 days (Venti, personal communication), respectively.
**2.3 Autonomous observing capabilities and dataset**
The autonomous capability of the Ma$p$CO$_2$ buoy provides continuous 3-hour measurements of both
$CO_{2,air}$ and $CO_{2,sw}$ mole fraction (xCO$_{2,air}$ and xCO$_{2,sw}$). These are converted to $p$CO$_2$ with total
uncertainties of <1 µatm and <2 µatm, respectively (Table 1; Sutton et al., 2014a). The buoy is equipped
with a seawater-gas equilibrator, reference gas standard, an infrared gas analyzer, Seabird 16/37
conductivity and temperature recorder, and a Sunburst$^{TM}$ SAMI-pH system located at about 1 m depth.
Details of the Ma$p$CO$_2$ instrument calibration and data Quality Assurance and Quality Control (QA/QC)
processes are described in Sutton et al. (2014a, 2014b). Final measurements used for the analyses cover
from January 2009 to January 2017.

The Ma$p$CO$_2$ buoys also have an internal Maxtec$^{TM}$ $O_2$ sensor (MAX-250+) located inside the $CO_2$
electronics tube (downstream of the infrared gas analyzer sensor) to measure percent of oxygen in air (%
$O_2$, ± 3%). These sensors are part of the observational network of all PMEL $CO_2$ buoys, but only intended
as a diagnostic tool. The sealed compartment that houses the sensor provides protection from seawater



and the effects of marine biofouling, and excessive temperature, humidity, and pressure changes. Despite
their diagnostic purpose, they have been shown to offer some utility for studying ocean biological
variability (e.g., Xue et al., 2016) even though measurements from the MAX-250+ may not be ideal due
to slow $O_2$ equilibration response times (Sutton et al., 2014a). Nevertheless, in an effort to achieve more
coherent and usable long-term estimates of $O_2$ concentrations, we obtain reasonable results when we post-
calibrate the MAX-250+ using early deployment data from the Aanderraa™ $O_2$ optode (Aanderaa 4775,
accuracy of $\pm 5$ %).

The $O_2$ data obtained by the optode sensor tends to be unreliable for extended deployments of over 6
months as a consequence of biofouling and subsequent sensor drift. Therefore, we proceed under the
assumption that they are accurate within specifications for a time period between calibration and prior to
any biofouling. The MAX-250+ sensor bias correction was derived from four optode deployments from
2011 to January 2017, and 20 *in situ* bottle Winkler $O_2$ (Winkler, 1888) measurements collected and
analyzed at the CARICOOS lab between 2015 and 2016. The $O_2$ optode measurements were salinity
compensated and an internal QA/QC process was applied where the first $O_2$ measurements were removed
(considered as the flush to purge residual water and bubbles in the flow lines) and the last two of each
cycle averaged. Seawater surface concentration (mmol m$^{-3}$) is calculated using the $O_2$ solubility (Garcia
and Gordon, 1992) as a function of SST and SSS. Linear regression analyses were performed at daily and
seasonal time scales (see supplemental material, Fig.S1 and S2) and the best fit ($r^2 = 0.90$, RMSE = 3, n
= 40) was found using the daily averages of the first 40 days after the first deployments, a time interval
whereby we judge the data accurate. We corrected the MAX-250+ measurements using the offset and
slope (Fig.S1). We also found the corrected $O_2$ measurements agreed reasonably well with the Winkler
$O_2$ determinations ($\pm$ 6.85 mmol m$^{-3}$).

Hourly wind measurements were taken from the nearby NOAA Integrated Coral Observing Network
(ICON) station. This station was maintained as an instrumented Coral Reef Early Warning System
(CREWS) station from 2006 to 2013 at the Media Luna middle-shelf reef (Fig.2-b), located 1.8 km south
from the Ma$p$CO$_2$ buoy. In 2015, CARICOOS established a meteorological station at the nearby Marine



Lab (Fig.2-b), 1.5 km from the Ma$p$CO₂ buoy. Wind speed from the ICON and CARICOOS stations were
measured at 6.5 m and 7.8 m height, respectively. Wind speeds were normalized to a wind speed at 10 m
height according to Hsu et al. (1994) and averaged to every 3 hours. Wind data is taken from this station
from 2015 to 2017. Wind data gaps were filled using a climatological curve created with these two
datasets. This method was preferred over the use of buoys far from the near-shore study site or satellite
wind measurements, as their use tends to overestimate the gas air-sea exchange due their low temporal
coverage and limited near-shore spatial coverage (e.g., Jiang et al., 2008). We believe that the coastal
topographic setting plays a role in setting up diurnal wind patterns, which are not captured by the remote
buoys or the satellite-derived wind speed data.
**2.4 *In situ* geochemical surveys**
Open ocean dynamics in carbonate chemistry are reasonably well constrained and often characterized
based on TA-salinity and $p$CO₂-temperature relationships (Lee et al., 2006; Gledhill et al., 2008).
However, in near-shore environments benthic and coastal processes can convolute these relationships.
Therefore, the autonomous observations were validated and supplemented on a weekly (2009-2013) and
bi-weekly (2013-2017) basis through direct measurements of potentiometric TA and spectrophotometric
pH. Phosphate and silicate concentrations were collected at the buoy site and measured according to
Strickland and Parson (1972) six times over the period from 2009 to 2011 in January, February, March,
May, November, and December. Details about the collection and analyses procedures of these *in situ*
measurements are available in supplemental material, S2.

The average difference between $p$CO$_{2,sw}$ buoy measurements and $p$CO$_{2,sw}$ calculated from *in situ* bottle
samples was <0.5 µatm. The average difference between SAMI-pH and spectrophotometric pH
measurements was 0.005. This data comparability, achieved with widely differing methods, demonstrates
the robust quality of our analytical procedures and resulting data. *In situ* bottle measurements helped with
the determination of site-specific algorithms for TA and allowed estimates of DIC and $p$CO$_{2,sw}$ from *in*
*situ* bottle samples (described in Sect. 2.6.4). These data sets were also used to investigate factors



controlling the $pCO_{2,sw}$ dynamics at the reef station using a 1-D mass conservation model (described in
Sect. 2.6).

## 2.5 First-order derivations of TA and calculation of carbonic acid system

TA algorithms based on seawater surface salinity (SSS) and temperature (SST) break down for inshore
waters subject to contributions from multiple freshwater end-members and $CO_3^{2-}$ ion variability during
calcification and dilution processes (Lee et al., 2006), requiring site-specific relationships to be
determined. At the Enrique forereef, *in situ* bottle TA measurements showed a moderate ($r^2 = 0.42$), but
robust ($p < 0.0001$, $n = 547$) correlation to SSS (see supplemental Fig.S3). This was improved with the
weak ($r^2 = 0.27$), but significant ($p < 0.0001$, $n = 548$) negative correlation with SST. The resultant
multivariate linear relationship between TA with SSS and SST produced an $r^2 = 0.55$ and $p<0.001$ (n=
547, RMSE = 30). This was used to model TA $\pm 30$ µmol kg$^{-1}$ at the reef (TA$_{reef}$):

$$TA_{reef} = 43.2(\pm 0.08) \times SSS - 12.5\,(\pm 0.07) \times SST + 1118\,(\pm 2.10) \qquad (5)$$
where, the $\pm$ is the standard error of the coefficients.

The reef $CO_2$ – carbonic system was fully solved using TA$_{reef}$ and the $pCO_{2,sw}$, along with SSS, SST, and
pressure (1 db – pressure at 1 meter) from the buoy measurements. The MATLAB program $CO_2$SYS (van
Heuven et al., 2011), applying the dissociation constants for K$_1$ and K$_2$ of Lueker et al. (2000) and for
K$_{HSO4^-}$ from Dickson (1990), was used.

## 2.6 Diagnostic mass budget model

A 1-D mass budget model was used to calculate the net change in $pCO_{2,sw}$ based on contributions from
the processes controlling the variability of $pCO_{2,sw}$ at the surface (e.g., Gruber et al., 1998; Shadwick et
al., 2011; Fassbender et al., 2016; Xue et al., 2016). The model is assumed to provide an integrated
assessment of biogeochemical variability throughout the well mixed water column as the water mass
flows in over the reef-shelf platform from the open ocean end member. The net changes in observed
$pCO_{2,sw}$ ($\delta pCO_{2OBS}$) were calculated from mass the balance as given by:




$$\frac{\delta pCO_{2OBS}}{dt} = \frac{\partial pCO_{2\,SOL}}{\partial t} + \frac{\partial pCO_{2\,AIR-SEA\,EX}}{\partial t} + \frac{\partial pCO_{2\,HOR\,MIX}}{\partial t} + \frac{\partial pCO_{2\,BIO}}{\partial t} \qquad (6)$$

where, the corresponding daily $\delta pCO_{2OBS}$ are from the partial changes ($\partial$) of gas solubility as a function
of SST and SSS (SOL), air-sea exchange (AIR-SEA EX), horizontal mixing processes (HOR MIX), and
biological activity (BIO). All the individual parameters used the observed daily means to avoid the effects
of high frequency (<24 hr) processes (e.g., tides and diurnal biological activity). The data were binned
within representative Julian days to create a composite year for each day from 2009 to 2017 (details in
Sect. 2.9).
**2.6.1 Thermodynamic variability**
The thermodynamic variability ($\partial pCO_{2SOL}$) was imposed on the system using daily-observed changes in
SST and SSS using the $CO_2SYS$ program. We preferred this method over the temperature only
dependence coefficient ($0.0423°C^{-1}$) by Takahashi et al. (1993) as both SSS and SST impart
thermodynamic variability in this region and the SST distributions are much different with respect to the
waters on which this dependence was derived.
**2.6.2 Variation by physical transport**
The physical transport attributable to horizontal transport via advection ($\partial pCO_{2HOR\,MIX}$) was characterized
empirically using SSS changes with assumed conservative mixing of TA and DIC between the reef and
open ocean (e.g., Xue et al., 2016). Changes due to vertical mixing are neglected and the mixed layer is
assumed to extend to the bottom given the small variations on SSS and SST, from the CTD casts
performed for the *in situ* geochemical surveys, and shallowness of the site (3 m). In addition, three tidbit
temperature loggers were deployed one year long from January through December 2015 at different
depths along the buoy assembly, showing no thermal stratification in the area. The seasonal change in
SSS due to the mean potential evapotranspiration to precipitation rate is assumed to be small (<0.1%) and
hence neglected. Mixing due to tides is ignored due to the limited diurnal tidal range (<0.25m) in the area.



*In situ* bottle TA and DIC samples were obtained from seasonal cruises around the Caribbean Region
(details in supplemental material, S3), including three cruises to the Caribbean Time Series station (CaTS,
Fig.2-b). The linear relationship between TA and SSS measurements (n= 237, r2 = 0.99, *p*<0.001, RMSE
= 4.79) was derived to model TA (TA$_{ocean}$). To model DIC (DIC$_{ocean}$) a linear relationship (n= 220, $r^2$ =
0.93, *p* <0.001, RMSE = 12) with SSS and SST was derived. We used the observed daily change in SSS
at the buoy to estimate daily changes of TA and DIC from the conservative mixing relationships as
follows:

$TA_{ocean}\ (\pm4.79)\ =\ 58.9\ (\pm0.02)\ \times\ SSS + (S_2 - S_1) + 237\ (\pm0.900)$     (7)
$DIC_{ocean}\ (\pm12)\ =\ 50.1\ (\pm0.25)\ \times\ SSS + (S_2 - S_1) - 4.13\ (\pm0.16)\ \times\ SST\ + 322\ (\pm12.5)$   (8)

where, the ± is the standard error of the coefficients or the RMSE of the derived quantity as applicable,
and S$_1$ and S$_2$ correspond to the salinity at time 1 and 2, respectively. Despite the large uncertainties
reported for the physical term using the mass conservation model approach (e.g. Shadwick et al., 2011;
Fassbender et al., 2016; Xue et al., 2016), the effect of such uncertainty on subsequent derivations is
minimal (details in Sect. 2.10).
**2.6.3 Air-Sea exchange**
The effects on $p$CO$_{2,sw}$ due to the physical transport through the air-sea CO$_2$ exchange ($\partial p$CO$_{2\,AIR\text{-}SEA\ EX}$)
is related to the DIC changes and  air-sea CO$_2$ flux (F$_{CO2}$). The DIC$_{AIR\text{-}SEA\ EX}$ ($\mu$mol kg$^{-1}$ day$^{-1}$) is estimated
via the change in mixed layer DIC inventory as:

$DIC_{AIR-SEA\ EX} = \dfrac{ks \times (fCO_{2,sw} - fCO_{2,air})}{h\rho}$     (9)

where, $f$CO$_{2,sw}$ - $f$CO$_{2,air}$ is the difference in atmospheric and seawater CO$_2$ fugacity ($f$CO$_2$) concentration,
calculated from the $x$CO$_{2,air}$ and $x$CO$_{2,sw}$ buoy measurements and converted according to best practices
(Dickson et al., 2007; SOP 5) using the virial coefficient of pure CO$_2$ and the virial coefficient of CO$_{2,air}$

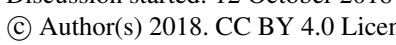


according to Weiss (1974). The $s$ (mol kg$^{-1}$ atm$^{-1}$) is the solubility of $CO_2$ per unit volume of seawater
(Weiss, 1974), $k$ (m s$^{-1}$) is the transfer velocity as a function of wind speed at 10 m above mean sea level,
$h$ is the mixed layer water depth (m), and $\rho$ the seawater density (kg m$^{-3}$). The transfer velocity-wind
speed relationship used is described by Wanninkhof (2014). As a convention in this paper, $O_2$ and $CO_2$
positive fluxes are from the ocean to the atmosphere.
**2.6.4 Modulation carbonate parameters by biological processes**
The biological processes affecting $p\text{CO}_{2,\text{sw}}$ ($\partial p\text{CO}_{2\text{BIO}}$) were estimated as the residual of the remainder of
the other terms on the mass conservation equation (Eq.6) to close the system. The changes on $\partial p\text{CO}_{2\text{BIO}}$
due to biological activity were defined as:

$\dfrac{\partial pCO_{2\,BIO}}{\partial t} = \dfrac{\partial pCO_{2\,NEP}}{\partial t} + \dfrac{\partial pCO_{2\,NEC}}{\partial t}$             (10)

where $p\text{CO}_{2\text{NEP}}$ is defined as the processes affecting $p\text{CO}_{2,\text{sw}}$ due to gross photosynthetic production and
community respiration and the $p\text{CO}_{2\text{NEC}}$ refers to the processes affecting $p\text{CO}_{2,\text{sw}}$ due to calcium carbonate
gross calcification and gross dissolution.

The Revelle Factor ($\beta$, $\partial\ln p\text{CO}_{2,\text{sw}}\,/\partial\ln\text{DIC}$) was used to convert changes in $\partial p\text{CO}_{2\text{BIO}}$ to changes in DIC
($\partial\text{DIC}_{\text{BIO}}$) using the relation between $p\text{CO}_{2,\text{sw}}$ and DIC as defined by Revelle and Suess (1957). The *in*
*situ* bottle pH and TA measurements were used to calculate the *in situ* bottle $p\text{CO}_{2,\text{sw}}$, the DIC, and the $\beta$.
The partial change in DIC due to $p\text{CO}_{2,\text{sw}}$ (DIC/$p\text{CO}_{2,\text{sw},}$, $\mu$mol kg$^{-1}$ $\mu$atm$^{-1}$) was related to $\beta$. This was
linearly related to changes in SST (n = 467, r$^2$ = 0.86, p <0.001, RMSE = 1.02; Fig.3). The $\partial p\text{CO}_{2\text{BIO}}$ was
converted to $\partial\text{DIC}_{\text{BIO}}$ according to the following relationship:

$\dfrac{\partial DIC_{BIO}}{\partial t} = \dfrac{\left(\dfrac{DIC}{pCO_2} \times \dfrac{\partial pCO_{2\,BIO}}{\partial t}\right)}{\beta}$             (11)





### 2.7 Net Ecosystem Production (NEP) and Net Ecosystem Calcification (NEC) rates

The difference between gross ecosystem photosynthetic production and respiration (NEP) was addressed using the observed $O_2$ ($O_{2OBS}$) measurements and the net $O_2$ air-sea flux ($F_{O2}$). The net change in $O_2$ due to organic production is defined as:

$$\frac{\partial O_{2NEP}}{\partial t} = \frac{\partial O_{2OBS}}{\partial t} - \frac{\partial O_{2GAS}}{\partial t} \tag{12}$$

where, $\partial O_{2OBS}$ (mmol m$^{-3}$) are the $O_{2OBS}$ changes and the $\partial O_{2GAS}$ (mmol m$^{-2}$ day$^{-1}$) are the $O_2$ changes due to $F_{O2}$. The $O_2$ fluxes were calculated using the transfer velocity (details in Sect. 2.6.3) and corrected by the bubble flux injection and the bubble flux exchange (Manning and Nicholson, 2016). The changes on $O_{2NEP}$ were converted to those of carbon using the Redfield carbon to oxygen molar ratio of 106:138 (close to the observed $O_2$ to DIC ratio of 1.1 at Enrique reef) and integrated over the mixed layer.

The NEC was estimated using equations (11) and (12) as follows:

$$\frac{\partial DIC_{NEC}}{\partial t} = \frac{\partial DIC_{BIO}}{\partial t} - \frac{\partial O_{2NEP}}{\partial t} \tag{13}$$

### 2.8 Ecosystem free energy budget

We recast our monthly $\Omega_{arag}$ climatology in terms of free energy by applying equation 4 and the equations detailed in Morse and Arvidson (2002) for three common mineral phases relevant to coral reef ecosystems: aragonite, 10% MgCO$_3$, and 15% MgCO$_3$. From the monthly climatological $\Omega_{arag}$ values generated from CO$_2$SYS we first derived the ion concentration product (*ICP*) using the stoichiometric solubility product (K*$_{SP}$, Eq.3) and subsequently converted it to *IAP* using the ratio of the thermodynamic to stoichiometric solubility products according to:

$$IAP = k_x \Omega_x \tag{14}$$






where, $k_x$ is the thermodynamic solubility product for $x$ calcium carbonate mineral phase (Plummer et al.,

461 1978).

**2.9 Constructing an annual climatology**

We use the average daily observations from 2009 to 2017 to construct the annual climatology for each
parameter and derived products to examine the seasonal cycles. The composite year was constructed by
binning the data within the representative Julian day. The time series was compiled into two distinct
seasons based on the weather in Puerto Rico. The dry and cool season extends from October to March,
considered winter. The rainy, humid and hot period runs from April to September and is defined as
summer. The seasonal variability was computed using the peak-to-peak amplitude. Details about how the
gaps were filled are described in the supplemental material, S4.

**2.10 Error assessments**

Model errors for the mass budget model variables (Eq. 6), NEP, and NEC were estimated using Monte
Carlo simulations. The same approach was used to estimate the uncertainties of $TA_{reef}$, $DIC_{reef}$, $DIC_{AIR-SEA\ EX}$, $DIC_{BIO}$, $\Omega_{arag}$, and $\beta$ linear regression coefficients. Prior to the Monte Carlo simulations, a
normality check was performed using the Kolmogorov-Smirnov test on each of the model variables to
verify that the data were normally distributed. Random normal distributions of each model variables
(Table 1) were generated using MATLAB. Sampling was repeated 1,000 times to establish the final
uncertainty, mean, and standard deviation for each simulation. Details about the error analyses are
describe in the supplemental material, S5.

The uncertainties associated with the buoy observations are relatively small compared with the
uncertainties in TA derived from the conservative mixing with SSS and SST (Eq. 5). This error is large
relative to the daily TA changes (which were typically small) and arises largely from non-conservative
biological processes, involved with net calcification and net dissolution, which cannot be captured in our
linear model. However, we demonstrate that TA uncertainties from the conservative mixing model are



less than the combined uncertainties on the TA anomaly technique from the seawater residence time and
depth (Venti et al., 2012; Falter et al., 2013; Teneva et al., 2013; Courtney et al., 2016). The 1-D mass
balance approach avoids the errors of the TA-anomaly method and provides accurate estimates of the
physical and biological processes. However, better constraint of the $TA_{reef}$ would certainly help reduce
uncertainties in NEC.

**2.11 Biogeochemical footprint**

To better constrain the uncertainty in the horizontal advection term, we used underway measurements of
SSS, SST, $O_2$, $pCO_{2,sw}$ collected in November 2016 and March 2017 that were intended to assess the
spatial variability around the Ma$p$CO$_2$ buoy. These surveys extended from the buoy site to >1km
offshore/inshore and incorporated averaged column velocity profiles at the buoy site. The maximum
$pCO_{2,sw}$ change observed during these surveys was $< \pm 0.5$ µatm day$^{-1}$ assuming the maximum observed
current speed (7 cm s$^{-1}$) recorded. Our estimated uncertainty of $pCO_2$ ($\pm 0.5$ µatm day$^{-1}$) is based on this
observation.  Considering a unidirectional ~7 cm sec$^{-1}$ current, a daily mean parameter could be
influenced by waters up to ~6 km from the buoy. In actuality, the length scale would be much less
considering diurnal changes in tides and wind direction. Further, gradients in carbonate parameters
between the different reef sites, seagrass, mangrove channels, sand, and offshore waters tend to be small
relative to the temporal changes addressed in this study. Based on these results and Enrique's
physicochemical characteristics (e.g., currents, winds, residence times, and SSS changes), we assume that
the local river inputs are small and that the forcing from offshore waters occurs at time scales much longer
than the reef residence time. Nevertheless, it is important to note that our estimates are not solely
influenced by a single reef community and instead are representative of the broader reef-shelf complex
system.



# 3 Results

## 3.1 Seasonal variability of the carbonate chemistry

On an annual scale, the Enrique reef experiences a seasonal SST daily average variations of about 4 ˚C (Table 2) with a maximum 30.2˚C during the summer (September) and a minimum of 26.6˚C during the winter (January). The SST at the reef is about 1 ˚C warmer and 1 salinity unit fresher than the offshore station CaTS. The high temperatures (>30 ˚C) between August and October coincide with low salinities (Fig.4-a). The SSS seasonal change is about 2 units with a maximum during the summer (April) and a minimum during the winter (November). Increased local rainfall causes recurrent decrease in SSS during the months of June through December. The $DIC_{reef}$ and $TA_{reef}$ show similar SSS seasonal patterns with maxima in April and minima between September and November (Fig.4-b). The DIC shows a smaller seasonal amplitude (~70 $\mu mol\ kg^{-1}$) compared with TA (~100 $\mu mol\ kg^{-1}$). Average phosphate and silicate concentrations were $0.032 \pm 0.018\ \mu mol\ L^{-1}$ and $1.83 \pm 0.277\ \mu mol\ L^{-1}$, respectively. The TA changes attributable to these inorganic nutrients at these concentrations are negligible.

Enrique forereef, like many other reefs, is a persistent source of $CO_2$ to the atmosphere ($2.04 \pm 2.13$ mmol $CO_2\ m^{-2}\ day^{-1}$) with a minimum during the winter and a maximum during the summer (Fig.4-c, Table 2). The seasonal amplitude of the $F_{CO2}$ is about 7 mmol $CO_2\ m^{-2}\ day^{-1}$ with an annual mean of $0.75 \pm 0.78$ mol $CO_2\ m^{-2}\ year^{-1}$. Conversely, the maximum net $F_{O2}$ outgassing (14 mmol $O_2\ m^{-2}\ day^{-1}$) from the ocean to the air occurs during the winter. During the summer, the system is a net sink with a maximum of about -90 mmol $O_2\ m^{-2}\ day^{-1}$ (Fig.4-c, Table 2). Average daily wind speed ranges from 3 m $s^{-1}$ to 6 m $s^{-1}$ and prevailing wind direction from the east and southeast, with little seasonal variation (Fig.4-f, Table 2). The injection of bubbles represents <2 % of the total $O_2$ flux variation at the site.

Figure 4-d shows the $pCO_{2,sw}$, $pCO_{2,air}$, and $O_2$ dynamics for the site. Maximum seawater $pCO_{2,sw}$ values were observed from August through October (late summer) while minima occurs in February when the surface $pCO_{2,sw}$ decreases to near atmospheric equilibrium (~390 $\mu atm$). The $pCO_{2,sw}$ seasonal amplitude is about 70 $\mu atm$ (Table 2). In contrast, $pCO_{2,air}$ fluctuations were modest (13 $\mu atm$) and similar to regional



$CO_{2,air}$ values reported (Conway et al., 1994; Masarie and Tans, 1995). Maximum $CO_{2,air}$ concentrations
are observed from January to May (late winter) and minimum from June through October (late summer).
The $O_2$ minimum (154 $\mu$mol kg$^{-1}$) occurs during the late summer when the $pCO_{2,sw}$ is at a maximum. The
annual range of $O_2$ concentration is about 38 $\mu$mol kg$^{-1}$ with a maximum (192 $\mu$mol kg$^{-1}$) during the
winter.

The pH and $\Omega_{arag}$ show similar patterns with minimum values from August to November (pH <8 and $\Omega_{arag}$
<3.5) and maximum from March to May (Table 2). The mean pH is $8.02 \pm 0.01$ with an annual amplitude
of 0.07 units. Mean reef pH conditions during the late summer are ~17 % lower ("more acidic") when
compared with the winter (Fig.4-e). The mean $\Omega_{arag}$ is $3.59 \pm 0.07$ with an annual amplitude of 0.3 units
(Table 2). These results are comparable with the seasonal results on Sutton et al., (2016) using the same
data set from December 2012 to December 2014.

Results from the 1-D mass balance model results shows that the increase in $\partial pCO_{2SOL}$ from May to mid-
October can increase water column $pCO_{2,sw}$ up to  about 20 $\mu$atm (Fig.5). During the same season, the
air-sea exchange drives the $pCO_{2,sw}$ up to about 2 $\mu$atm decreasing DIC in the water column by only ~5
$\mu$mol kg$^{-1}$. From July through December, the $\partial pCO_{2BIO}$ drives up the water column $pCO_{2,sw}$ by about 8
$\mu$atm. The $\partial pCO_{2HOR\ MIX}$ does not show a significant seasonal change and we find its contribution to be
negligible (<1 $\mu$atm) throughout the year.
**4 Discussion**
**4.1 Net Ecosystem Processes**
High TA to DIC slope (>0.5) from the *in situ* surveys is observed in Enrique forereef indicating the strong
influence of net ecosystem metabolic processes over DIC and TA (Fig.6). High TA to DIC slopes have
been used to qualitatively suggest that the system shows net calcification. However, despite Enrique's
high TA to DIC slope (1.1) calculated in this study and compared with other reefs areas in the Atlantic
and Pacific by Cyronak et al., (2018), the excess of nTA and nDIC relative to adjacent Caribbean oceanic



values indicate that dissolution and respiration are dominant processes. Figure 6 shows how nTA and
nDIC values are greater than the nTA and nDIC oceanic mean conditions (TA and DIC excess) indicating
net ecosystem heterotrophy and carbonate dissolution conditions. It also shows that photosynthesis and
calcification slightly dominate at some point in time when the TA and DIC values are below the oceanic
mean conditions (TA and DIC consumption). This suggests that the $CO_2$ production during respiration
could play an important role on pH and $\Omega_{arag}$ dynamics at seasonal time scales.

**4.2 Physical and Biological drivers of $p$CO$_{2,sw}$**

The 1-D mass balance model shows that throughout the year, solubility and biological changes are the
dominant processes on the water column $p$CO$_{2,sw}$ dynamics at Enrique forereef (Fig.5). The solubility
effects are driven by the small SST seasonal variation (4˚C), primarily due to the smaller amplitude in
regional air temperatures and therefore, net atmospheric heat flux. The increase in SST during the early
summer is the main driver increasing water column $p$CO$_{2,sw}$ and F$_{CO2}$ from July to October. The SSS
seasonal change is associated with the regional SSS dynamics from the extension of Amazon and Orinoco
river plumes into the northeastern Caribbean (Corredor and Morell, 2001). The cooling and lower DIC
pool from the freshwater influx of the regional river plumes have the opposite effect driving the $p$CO$_{2,sw}$
down. Biological processes (NEC and NEP) exert the second most important control on the $p$CO$_{2,sw}$ and
F$_{CO2}$ dynamics at this site on an annual scale (details in the next sections). The physical processes, i.e.
horizontal transport and the removal of $CO_2$ from the mixed layer via air-sea exchange, play a small role
on $p$CO$_{2,sw}$ .

**4.3 Net ecosystem metabolism rates**

On an annual basis, the NEP is about 69.5 ± 9 g C m$^{-2}$ year$^{-1}$ and NEC is about -581.9 ± 80 g CaCO$_3$ m$^{-2}$
year$^{-1}$) at La Parguera reef shelf system. Under present day conditions the MapCO$_2$ buoy shows that both,
dissolution and heterotrophic conditions persists for about 76 % (278 days) of the year in La Parguera
reef system. This is longer than the time interval of dissolution found in the Pacific (Yates and Halley,
2006) and the Western Atlantic (Muehllehner et al., 2016). Our seasonal net ecosystem metabolic
measurements are integrated over a longer time scales and spatial extent making difficult to compare with



studies that are based on short-time reef community diurnal experiments. However, at the uncertainties
reported by other methods (e.g., TA-anomaly) this study may not be particularly anomalous in terms of
dissolution considering that most of these studies are focused on a specific community. One of the caveats
of the method is that is based on ecosystem processes in a near-shore reef zone that includes TA and DIC
fluxes that do not originate from the reef. This could mask the coral reef biological signature. Additional
studies would be required to evaluate and constraining these processes.

Based on our results, respiration is the major process dominating organic carbon metabolism at the site.
Daily rates of NEP reef ranged from -11 to 67 mmol C m$^{-2}$ d$^{-1}$ (Fig.7-a). Negative NEP (NEP < 0) values
represent net productivity (autotrophy) while positive (NEP > 0) values represent net respiration
(heterotrophy). Results on McGillis et al., 2009; McGillis et al., 2011 from the gradient flux (CROSS)
and enclosure (SHARQ) methods estimated NEP from March 27$^{th}$ - 29$^{th}$ of 2009 at Enrique reef of about
43.1 C mmol m$^{-2}$ day$^{-1}$ and 60.3 C mmol m$^{-2}$ day$^{-1}$, respectively. Our results show an average of 26.3
mmol C m$^{-2}$ day$^{-1}$. These rates are similar, but slightly less than the results from CROSS and SHARQ.
We believe the discrepancy between estimates is mainly because our model uses measurements taken at
the near-surface and thus integrates values over the water column and at some horizontal length scale. In
contrast, the CROSS and SHARQ systems measurements are restricted to the benthic boundary.
However, all methods showed net heterotrophic conditions during the studied days.

The seasonal rates of NEC range from -58 to 12 mmol CaCO$_3$ m$^{-2}$ d$^{-1}$ (Fig.7-b). Positive NEC values
represent net calcification (NEC > 0) while negative values represent net dissolution (NEC < 0). The
average dissolution rate of -22.6 mmol m$^{-2}$ day$^{-1}$ (-2.26 g m$^{-2}$ day$^{-1}$), is at the higher end of rates reported
for a coral reef system. Experimental studies have demonstrated that NEC rates are controlled by
community composition and environmental conditions (e.g., Langdon and Atkinson, 2005). The high
dissolution rates at Enrique forereef may be a function of the area occupied by reef-building corals at the
site (~10%). Large reductions in live coral cover have occurred across the Caribbean (Gardner et al.,
2003). This has caused a shift of reef community structure as most reefs are no longer being dominated
by scleractinian corals and this has led to a decrease in calcification rates (Perry et al., 2013). Since our





model tracks the temporal behavior of TA, our high net dissolution rates could be influenced by TA
subsidies not accounted for in this study. This includes biotic and abiotic processes at the carbonate
sediments and coral rubble that cover around 35% of the Enrique forereef benthic area. For example, net
dissolution rates have been reported for soft bottom/sediment communities (Boucher et al., 1998) and
sand (Yates and Halley, 2006) at a rate of -2.4 mmol $m^{-2} d^{-1}$ and -3.2 mmol $m^{-2} d^{-1}$, respectively. Another
source of TA may arise via anoxic processes in the benthic community (e.g., sulfate reduction).

We consider that the 1-D mass balance approach using buoy $pCO_{2,sw}$ and $O_2$ observations produces
similar NEC and NEP values as the "slack water" (e.g., incubations or mesocosms) and non-enclosure
approaches (e.g., gradient flux, Eularian, Lagrangian and control volume of the seawater overlying the
benthic community). Our method also addresses their combined influence of benthic and water column
processes as well as the effects of $CaCO_3$ dissolution on net ecosystem processes within the mid-shelf
reef areas of La Parguera Marine Reserve. It must also be pointed out that our mass balance approach
does not provide absolute values for NEP and NEC, but rather provides a climatological view of seasonal
changes in the balance of net heterotrophy and net autotrophy and net calcification and net dissolution.
We submit that this could prove more useful in terms of coral reef management as a tool to monitor the
reef system health across different temporal scales.

Further studies to validate this method against other methods that can capture the seasonal variability on
NEP and NEC at the site are needed. Validation of the NEC and NEP estimates from this method, either
directly or from nutrient or oxygen inventories, along with an understanding of hydrodynamics are needed
to constraint the effective footprint of the buoy measurements and to better discern community responses.
These additional assessments are necessary to predict the rates and magnitude of OA in near-shore reef
ecosystems.
**4.4 Seasonal dynamics and physical drivers of net ecosystem metabolism**
The seasonal cycle of NEP indicates that most of the net organic carbon fixation occurs during the winter
(Fig.7-a). This resulted in $pCO_2$ values close to atmospheric equilibrium. At the beginning of the winter



season (December) the system begins to switch from being heterotrophic to autotrophic, indicating that community photosynthetic production in the winter is larger than the respiration (from December to March). While production is an important process throughout the winter, respiration, particularly from June and October, generates a local source of DIC to the system.

Gray et al. (2012) observed the same seasonal over-saturation of $pCO_{2,sw}$ at nearby Media Luna reef during the summer and fall months from 2007 to 2008 and suggested that net heterotrophy was driven by remineralization of mangrove-derived organic carbon inputs. We speculate that the main drivers of the carbon excesses from mid-April to mid-December include a combination of exposure to exogenous organic carbon sources from inshore mangrove channels and local red mangroves stands. The shoreline of La Parguera urban area is influenced by increased nutrients and turbidity from terrigenous sources (Otero and Carbery, 2005), which would also contain dissolved and particulate organic matter. Such fluxes have been reported as primary threats to near-shore coral reef ecosystems (Garcia-Sais et al., 2008). Observed heterotrophy also coincides with the wet season and the seasonal decrease in SSS caused by the remote influx of freshwater originating from the Orinoco and Amazon River plumes (Corredor and Morell, 2001). Inputs of coastal dissolved organic matter (DOM) and particulate organic matter (POM) from these remote sources may significantly contribute. Continued investigation on coastal organic matter components from local and remote sources will provide additional information on the sources of this excess carbon.

We show that net calcification in Enrique forereef is coincident with maximum autotrophy, corresponding with high $\Omega_{arag}$ and pH. It appears that the increase on net productivity and SSS from January to March may enhance $\Omega_{arag}$ providing favorable conditions for calcification. This is consistent with the coral reef ecosystem feedback hypothesis (CREF, Bates et al., 2009), where increasing $\Omega_{arag}$ is stimulated by an increase in net productivity. The combined seasonality in solubility (SST and SSS), and net ecosystem metabolic processes at the site act synergistically to exacerbate OA by depressing surface $\Omega_{arag}$ and calcification from April to December. Therefore, net ecosystem metabolic processes provide positive feedback driving $\Omega_{arag}$ down (Fig.8). Namely, continued release of $CO_2$ due to respiration from April to



December keep $\Omega_{arag}$ and pH low, driving metabolic dissolution (Anderson and Gledhill, 2013) which in
turn increases $p\mathrm{CO}_2$ and lowering pH.

The seasonal correlation of $\Omega_{arag}$ and NEC with SST and SSS is shown in Figure 9. The transition of NEC
from net calcifying to net dissolving occurs in April when the seawater $\Omega_{arag}$ is <3.6, SST is > 27, and
SSS is < 35 (Fig. 9). The biological effects exhibit hysteresis in NEC and $\Omega_{arag}$, while the horizontal range
shows the thermodynamic effects on $\Omega_{arag}$. This is a characteristic of dynamic systems when multiple
drivers can exist for the same set of parameter values (e.g., NEC or NEP). These seasonal shifts can be
due to the combined synergy of local terms (thermodynamics, nutrients, light availability and water flow)
and regional (river inputs) processes that can influence reef metabolism (e.g., Yeakel et al., 2015; Bates,
2017). Enrique reef shows that when thermodynamic effects increase and $\Omega_{arag}$ is high (>seasonal mean)
the system starts shifting to net calcification and autotrophic conditions (winter: January to March). As
the SSS decreases, the SST increases, and $\Omega_{arag}$ decreases below the seasonal mean. During this time the
conditions are net heterotrophic and net dissolution rates increase, relative to the values above the $\Omega_{arag}$
seasonal mean.  The underlying NEC- $\Omega_{arag}$ seasonal hysteresis reflect that $\Omega_{arag}$ is not the main driver of
NEC and that NEP and SST seems to control carbonate precipitation at this site.

**4.4 Net dissolutional conditions**
The average net dissolution at Enrique is about 4 times more than calcification. On an annual basis, the
loss in carbonate mass observed in La Parguera from April to December is about $629 \pm 1.6$ g CaCO$_3$ m$^{-2}$
year$^{-1}$, a significant (t-test, $p$ <0.0001) fraction of the winter calcification rate ($40 \pm 0.28$ g CaCO$_3$ m$^{-2}$
year$^{-1}$). Considering only the entire forereef area of Enrique (0.0656 km$^2$; Zayas, 2011), this corresponds
to a loss of 45 tons of CaCO$_3$ vs 2.9 tons of CaCO$_3$ precipitated. These results indicate that La Parguera
shelf reef systems are losing CaCO$_3$ by the process of metabolic dissolution at a rate far faster than
production and has already crossed a threshold point for net positive carbonate production.



Enrique reef is well below the boundary point for where it has been shown that Caribbean reefs shift from
net accretion to net erosion (Perry et al., 2013, 2015). Critical calcification/dissolution threshold values
have been estimated to occur when $pCO_{2,sw}$ between the 450 - 1000 µatm and $\Omega_{arag}$ <3 (Langdon et al.,
2003, Yates and Halley, 2006; Silverman et al., 2007; Andersson et al. 2009; Shamberger et al., 2011;
Turk et al., 2015). These thresholds vary from one reef to another as a consequence of benthic community
composition, structural complexity (i.e., reactive surface area), variations in residence times (which effect
the time-space integration), light, hydrodynamics, and nutrient availability (Yates and Halley, 2006;
Silverman et al., 2007; Shamberger et al., 2011). Such thresholds are also dependent on the % of live
coral cover and the combination of erosional, dissolution, and bio-erosional processes at the reef (Perry
et al., 2013).

Our model estimates show the system exhibiting net $CaCO_3$ dissolution (NEC <0) on an annual basis,
although $\Omega_{arag}$ does not exceed the $\Omega_{arag}$ and $pCO_{2,sw}$ thresholds suggested by others.  Indeed, dissolution
rates at Enrique reef are higher than the levels projected for the 21st century (Eyre et al., 2018). Although,
most modern tropical coral reef systems currently reside well above potential critical thresholds to
maintain net reef calcification or accretion (Guinotte et al., 2003; Perry et al., 2013, 2015), near-shore
Caribbean coral reefs with low live coral cover could have reached these tipping points due to disease
and bleaching (linked to high SST), leading to low rates of NEC. For example, discrepancies are found
with the results from the census-based carbonate budget data collected at Enrique reef in August 2015
reported a mean net $CaCO_3$ production 1.04 (± 1.26) kg m$^{-2}$ year$^{-1}$ and an accretion rate of 0.55 ± 0.62
mm year$^{-1}$ (Perry et al., 2018).  This method suggests that the Enrique reef exhibited positive accretion,
albeit minimally and with high variability.  These budget surveys are not directly comparable to our NEC
rates, as they do not incorporate chemical dissolution and they are based on data obtained from a single
point observation in time. In addition, the census approach is based on the sum of the literature estimated
calcification rates by individual $CaCO_3$ producers whereas our approach integrated abiotic and biotic
processes over the entire reef system. Our current estimates are controlled by a combination of the open
ocean end member and biogeochemical and hydrodynamic processes on the reef shelf. The results from
these studies highlight the need to develop methods to better understand the metabolic processes and



seawater chemistry across functional, spatial, and temporal scales. Further refinement of our model is
required to better understand the carbonate chemistry variability at each functional community scale.

The highest rates of net dissolution occur during periods when rates of net heterotrophy are at their
highest. This corresponds to the periods when highly soluble minerals, cements, and organisms such as
crustose coralline algae exhibit net dissolution. An increase in the magnitude or time of occurrence of net
dissolution could become a serious threat to the formation of sediments, reefs, and other structures
composed of $CaCO_3$ (Eyre et al., 2014; 2018). It is clear, that under continued periods of net dissolution,
net reef accretion is jeopardized in the face of other climate stressors (e.g. coral bleaching and ocean
warming). Our model, coupled to the Ma$p$CO$_2$ data provides the ability to monitor and understand near-
shore carbonate chemistry variability to assess, forecast, and model the impacts of OA on near-shore reef
ecosystems. Prior to design of measures that may effectively ameliorate the impact of OA on tropical
shallow coral reefs ecosystems and allow us to improve current management strategies; we need to
develop a detailed understanding of the carbon cycle role in net ecosystem metabolic "performance". This
will aid in the identification and development of effective adaptation mitigation strategies that could
reduce local carbon inputs and decrease erosional processes.
**4.5 Free energy dynamics at La Parguera Near-Shore Reef ecosystems**
The use of $\Omega_{arag}$ as an index of coral health has perhaps been over-utilized in the OA literature to date,
with few instances of explicit acknowledgment of what the index truly represents nor the key assumptions
implied when adopting its use. Firstly, when considering the use of $\Omega_{arag}$ within the context of a coral reef
ecosystem, it should be appreciated that a coral reef comprises a heterogeneous composition of a range
of $CaCO_3$ minerals. While aragonite dominates, considerable amounts of magnesium carbonate with
variable ranges in magnesium content are also present. Furthermore, the stoichiometric solubility product
used by CO2SYS to generate the $\Omega_{arag}$ value is experimentally derived using abiotic aragonite while the
vast majority of $CaCO_3$ minerals deposited on coral reefs are biogenically derived. The behaviour of these
biogenically produced minerals is complex in seawater owing to the important role of kinetics so care
must be taken to avoid over-interpreting a reported saturation state with a complex marine system such



as the La Parguera (see review by Morse et al., 2007). Finally, it should be remembered that saturation
state is fundamentally related to the free energy of the system to drive the reaction which can be
capitalized by marine organisms. While corals have evolved a range of mechanisms to induce calcification
(at least adult life stages), these are energy intensive processes having to overcome the kinetic barriers to
calcification.  The less free energy available in the ambient environment to drive reaction, the more energy
corals will need to utilize for calcification.

The annual median free energy available to drive mineral precipitation for this Caribbean system are
given in Table 3. On the La Parguera shelf reef system, net dissolution would be favored for Mg-calcite
with >28 mol % $MgCO_3$. It is important to consider that the free energies shown here are based on
solubilities for synthetic mineral phases while biogenic phases are considerably more soluble (Busenberg
and Plummer, 1989). This would imply that the "true" free energies may be only a fifth that derived here
owing to their distinct differences in physical properties, composition, structure, and reactivity relative to
synthetic phases which are commonly assumed (Morse et al., 2007).

Adult marine calcifiers deploy a range of strategies to overcome this inhibition and drive the reaction at
rates that often depart significantly from abiotic processes. However, the initial energy available to drive
the reaction is reflected in the free energy pool of the ambient seawater. Whatever energy deficit may
exist between the ambient pool and that needed to induce nucleation is energy that the organism must
contribute itself. Due to the seasonal dynamics in free energy at Enrique reef, a coral must contribute an
additional ~ 85 cal mol$^{-1}$ $CaCO_3$ in the late fall relative to the spring regardless of the mineral phase being
considered (Fig. 10).

This seasonal dynamic is relatively minor in comparison with the loss of the free energy that has been
depleted from the system due to OA since the pre-industrial period, which has likely been greater than
155 cal mol$^{-1}$ $CaCO_3$.  By the end of the century under business-as-usual we can expect a loss ~337 cal
mol$^{-1}$ with respect to pre-industrial values, which will need to be compensated for by marine calcifiers in
this system if they are going to maintain calcification.





## 5 Conclusions

Net heterotrophy and net dissolution dominate over most of the year on the Enrique forereef and off-reef areas while net autotrophic conditions coupled with calcification dominated from only January to mid-April (winter). The seasonal hysteresis between NEC and $\Omega_{arag}$ suggest that $\Omega_{arag}$ is not the main driver of NEC at the site. The NEC dynamics are mainly controlled by NEP and SST and reflect the combination of the biogeochemical and hydrodynamic processes on the reef shelf from the open ocean waters. This carbon subsidy likely originated from nearby mangroves and autochthonous benthic processes with much of the dissolution generated within the sediments and reef framework. The Enrique forereef serves as a continuous net source of $CO_2$ to the atmosphere. This is largely driven by the seasonal cycle of SST. The thermodynamics and metabolic activity are the dominant processes influencing the $pCO_{2,sw}$ changes at Enrique forereef, with air-sea $CO_2$ gas exchange and advective processes having minor impacts on the carbonate chemistry over the 2009 - 2017 period.

Constraining the local near-reef variability in carbonate chemistry across diel, seasonal, and annual scales is an important step in determining potential biogeochemical thresholds to OA for specific reef environments. At seasonal timescales, the Enrique net reef community metabolism may also affect the reef's susceptibility to pressures from OA. It is likely that Enrique Reef and other nearby reefs in La Parguera are already experiencing prolonged periods of net dissolution driven by high respiration rates. Future projections of the net erosional processes (chemical and biological) can be anticipated, assuming that surface waters continue to increase in accordance with current trends in $CO_2$ and temperature. While these waters will remain supersaturated with respect to aragonite for at least several centuries, the most soluble carbonate mineral forms such as 14 mol % $MgCO_3$, found in Enrique sediments and the marine lithified (cements) in the coral reef framework, are likely to experience prolonged undersaturation on an annual basis. This also have implications on the coral energy required to maintain active calcification during periods of low free energy. Based on similarities in environmental characteristics, our results suggest that tropical Caribbean reef ecosystems are exhibiting periods of net dissolution.



Our results demonstrate that current sustained high temporal robust autonomous capabilities of buoyed
operational systems such as the La Parguera Ma$p$CO$_2$ buoy can be used to detect metabolic processes
sensitive to OA. Future research efforts should be directed to combine different techniques and *in situ*
methods for model validation in order to gain a better understanding of ecosystem changes across different
temporal scales. Our methods yield a valuable index of cumulative net ecosystem effects of the biological
processes affecting the water column over the reef.  It is also important to note that we use data from fixed
assets and that the methods may be scaled wherever similar data streams are found. Such results can be
used to establish critical baselines against which future comparisons can be made, thus enabling
evaluation of coral reef health and changes attributable to multiple stressors including OA. Further, high
frequency data provided by this and similar operational systems can be used to develop early warning
capabilities needed to identify and predict ecological community changes.
**Data availability**
The CO$_2$ buoy data are archived on the Ocean Carbon Data System (OCADS) at
https://www.nodc.noaa.gov/ocads/data/0117354.xml; Sutton et al., (2014b).
The data from the ICON station is archived on National Centers for Environmental Information (NCEI)
at https://www.nodc.noaa.gov/archive/arc0072/0124000/1.1/data/1-data/icon-lppr1-2013-data.csv;
Hendee (2014).
The discrete data are archived on the NCEI at
http://www.nodc.noaa.gov/oceanacidification/data/0145164.xml; Melendez et al. (2016).
The benthic community cover data are archived on the NCEI at https://data.nodc.noaa.gov/cgi-
bin/iso?id=gov.noaa.nodc:0157740; Manzello et al. (2017).
**Competing interests**
The authors declare that they have no conflict of interest.



## Acknowledgements

Support for this study was provided by the NOAA's Coral Reef Conservation Program through its AOAT project and the Ocean Acidification Program (OAP). UPRM participation in the latter is made possible through the IOOS partnership with CARICOOS program. We would like to give special thanks to Dr. Jorge Corredor for the opportunity of working with him and his work throughout the development of the AOAT. We also thank the staff and students of the UPRM Marine Sciences Department and UNH for field and lab assistance. We thank Helena Antoun, Belitza Brocco, Val Hensley and Erick García from UPRM for the assistance provided with the field and laboratory samples. We would like to thank Alyson Venti for providing the Be-7 sampling and residence time analyses. The authors are grateful for the constructive comments from Chris Hunt and Tyler Winsor. PMEL contribution number 4830.

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




**Table 1: MapCO$_2$ observations and model uncertainties.**

| Buoy Observations | Error ± | Method |
|---|---|---|
| Temperature (°C) | ± 0.01 | Sutton et al. (2014a) |
| Salinity (SSS) | ± 0.05 | Sutton et al. (2014a) |
| $p$CO$_{2sw}$ (μatm) | ± <2 | Sutton et al. (2014a) |
| O$_2$ | ± 3 % | See Section 2.3 |
| Model parameter | | |
| TA$_{reef}$ (μmol kg$^{-1}$) | ± 16 | Monte Carlo |
| DIC$_{reef}$ (μmol kg$^{-1}$) | ± 12 | Monte Carlo |
| Ω$_{arag}$ | ± 0.12 | Monte Carlo |
| CO$_2$ Flux (mmol m$^{-2}$ day$^{-1}$) | ± 0.72 | Sum of the squares |
| O$_2$ Flux (mmol m$^{-2}$ day$^{-1}$) | ± 2.04 | Monte Carlo |
| DIC$_{AIR-SEA EX}$ (μmol kg$^{-1}$) | ± 0.24 | Monte Carlo |
| DIC$_{BIO}$ (mmol m$^{-2}$) | ± 0.3 | Monte Carlo |
| Revelle Factor | ± 0.66 | Monte Carlo |
| $\partial$pCO$_{2,OBS}$ (μatm) | ± 1.1 | Monte Carlo |
| $\partial$pCO$_{2,SOL}$ (μatm) | ± 0.3 | Monte Carlo |
| $\partial$pCO$_{2,AIR-SEA EX}$ (μatm) | ± 0.1 | Monte Carlo |
| $\partial$pCO$_{2,HOR MIX}$ (μatm) | ± 0.5 | Gradient observations[*] |
| $\partial$pCO$_{2,BIO}$ (μatm) | ± 1.2 | Monte Carlo |
| NEP (mmol C m$^{-2}$ day$^{-1}$) | ± 2.08 | Monte Carlo |
| NEC (mmol CaCO$_3$ m$^{-2}$ day$^{-1}$) | ± 2.18 | Monte Carlo |

*refer to Sect. 2.6.2 for details









**Table 2: Summary of the reef station seasonal variability of the carbonate and oceanography parameters from 2009 to 2017. The (±)**
**is the standard deviation.**

| Parameter | Mean | Max | Min | Seasonal change |
|---|---|---|---|---|
| Temperature (°C) | 28.5 ± 0.41 | 30.2 | 26.6 | 3.62 |
| Salinity | 35.3 ± 0.26 | 36.1 | 34.4 | 1.63 |
| Wind speed (m s$^{-1}$) | 4.21 ± 1.76 | 6.38 | 2.36 | 4.01 |
| Residence Time (days)[*] | 15 ± 3 | 9 ± 2 (Jan) | 13 ± 2 (May) | 4 |
| DIC (µmol kg$^{-1}_{sw}$) | 1972 ± 13.9 | 2000 | 1931 | 69.3 |
| TA (µmol kg$^{-1}_{sw}$) | 2287 ± 13.7 | 2332 | 2231 | 101 |
| Revelle Factor | 9.25 ± 0.11 | 9.49 | 9.06 | 0.43 |
| $p$CO$_{2,sw}$ (µatm) | 423 ± 13.5 | 461 | 387 | 73.1 |
| $p$CO$_{2,air}$ (µatm) | 386 ± 8.52 | 391 | 378 | 12.6 |
| $\Delta p$CO$_{2,sea-air}$ (µatm) | 38.2 ± 14.2 | 80.8 | 0.72 | 80.1 |
| pH (total scale) | 8.02 ± 0.01 | 8.06 | 7.98 | 0.07 |
| $\Omega_{arag}$ | 3.59 ± 0.07 | 3.76 | 3.4 | 0.36 |
| O$_2$ (µmol kg$^{-1}$) | 176 ± 10.6 | 192 | 154 | 37.8 |
| F$_{CO2}$ (mmol m$^{-2}$ day$^{-1}$) | 2.04 ± 2.13 | 6.78 | 0.11 | 6.67 |
| F$_{O2}$ (mmol m$^{-2}$ day$^{-1}$) | -23.3 ± 22.5 | 14.2 | -86.6 | 72.4 |
| NEC (mmol m$^{-2}$ day$^{-1}$) | -18.1 ± 2.1 | 12.0 | -57.7 | 45.7 |
| NEP (mmol m$^{-2}$ day$^{-1}$) | 17.9 ± 2.2 | 66.8 | -10.9 | 55.9 |

[*]Reef water residence time relative to the offshore station located ~11 km from Enrique reef (for details, see Sect. 2.2).
**Table 3: Annual median Gibbs Free Energy (ΔG, cal mol$^{-1}$ CaCO$_3$) for a range of CaCO$_3$ mineral phases under ambient conditions**
**within the waters overlying Enrique reef.**

|  | Aragonite | Calcite | 5% MgCO$_3$ | 10% MgCO$_3$ | 13% MgCO$_3$ | 25% MgCO$_3$ | 30% MgCO$_3$ |
|---|---|---|---|---|---|---|---|
| ΔG (cal mol$^{-1}$ CaCO$_3$) | 754 | 1002 | 858 | 775 | 699 | 199 | -104 |






**Figure 1: Multi-annual time series of the Ma$p$CO$_2$ buoy (black line) and *in situ* (green dots) geochemical measurements, and**
**derived values from 2009 to 2017 at the forereef of Enrique reef. Final measurements used for the analyses cover from January**
**2009 to January 2017. The data from Feb 2017 to December 2017 is preliminary see:** www.pmel.noaa.gov/co2/story/La+Parguera**.**
**Sea surface observations of: a) temperature (Temp; $^0$C), b) salinity (Sal), c) Derived total alkalinity (TA; µmol kg$^{-1}$) as a function**
**of SST and SSS (Eq.5), d) dissolved inorganic carbon (DIC; µmol kg$^{-1}$) as a function of $p$CO$_2$ and TA, e) seawater and boundary**
**layer atmospheric $p$CO$_2$ ($p$CO$_{2,sw}$, $p$CO$_{2,air}$; µatm), f) air-sea gradient in $p$CO$_2$ ($\Delta p$CO$_2$; µatm), g) Derived pH as a function of $p$CO$_2$**
**and TA and pH measurements from the SAMI-pH system (orange line), and h) Derived oxygen using the MAX-250+ sensor (O$_2$;**
**µmol kg$^{-1}$). Data gaps filled climatological averages are highlighted in the gray shaded areas.**




**Figure 2: a) Map showing the town of Lajas (18.05° N, 67.05° W) and the Marine Protected Area of La Parguera located on the**
**southwest coast of Puerto Rico (orange box). b) The offshore sample stations (white dashed line) include the offshore station (17.87**
**N, -67.02 W) located ~11 km from Enrique reef and 1.6 km south (seaward) of the shelf-edge, and the Caribbean Time Series**
**station (CaTS, 17.36 °N, 67° W) at ~52 km offshore. c) The Ma$p$CO$_2$ buoy (17.95° N, 67.05° W) is located at the west-end of Enrique**
**(yellow dot) and over the forereef where the water depth is ~ 3 m. Red mangroves have colonized the emergent reef island**
**comprised by coral rubble. The dominant ocean current direction at the sea surface is towards the north-west.**



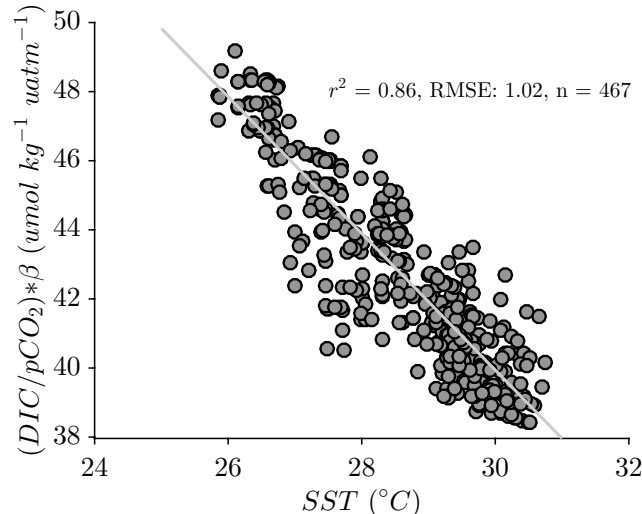


**Figure 3: Linear relationship between the fractional DIC (µmol kg$^{-1}$) and $p$CO$_{2,sw}$ (µatm$^{-1}$) to Revelle Factor (β) and SST (ºC). This linear relationship (y = SST × -2.02 (± 0.002) + 100 (± 0.063)) was used to compute ∂DIC$_{BIO}$ from ∂$p$CO$_{2BIO}$ according to equation 11.**






**Figure 4: Seasonal series of the buoy measurements, hydrographic properties (e.g., wind speed, fluxes) and derived carbonate**
**chemistry parameters (e.g., DIC, pH, $\Omega_{arag}$). The 3-hour observations from 2009 to 2017 were binned and averaged by day. Mean**
**(solid line) and standard deviation (shaded bounds) demonstrate the seasonality of sea surface a) temperature (Temp; $^0$C; orange)**
**and salinity (Sal; black), b) total alkalinity (TA; µmol kg$^{-1}$; orange) and dissolved inorganic carbon (DIC; µmol kg$^{-1}$; black), c) CO$_2$**
**(orange) and O$_2$ (black) flux (mmol m$^{-2}$ day$^{-1}$), d) post-corrected oxygen from the MAX-250+ sensor (O$_2$; mmol m$^{-2}$; orange),**
**seawater (black) and air (blue) $p$CO$_2$ ($p$CO$_{2,sw}$, $p$CO$_{2,air}$; µatm), e) $\Omega_{ARG}$ (orange) and pH (black) and f) air-sea gradient in $p$CO$_2$**
**($\Delta p$CO$_2$; µatm; orange) and wind speed (m s$^{-1}$; black).**



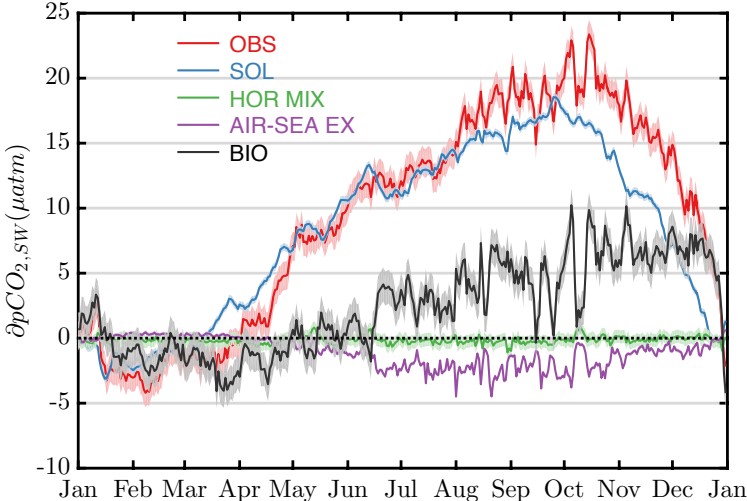


**Figure 5: Cumulative seasonal change in surface $p$CO$_{2,sw}$ ($\delta p$CO$_{2,sw}$; μatm) are based on contributions from thermodynamic, physical, and biological processes at Enrique forereef. The average (solid line) is determined from 2009 to 2017 data, with model uncertainties (shaded bounds, Table 1). Observed $p$CO$_2$ values (OBS; red), effects of temperature and salinity variability (SOL; blue), effects of horizontal mixing (HOR MIX; green), effects of air-sea exchange (AIR-SEA EX; purple) and effects of biological activity (BIO; black).**

326

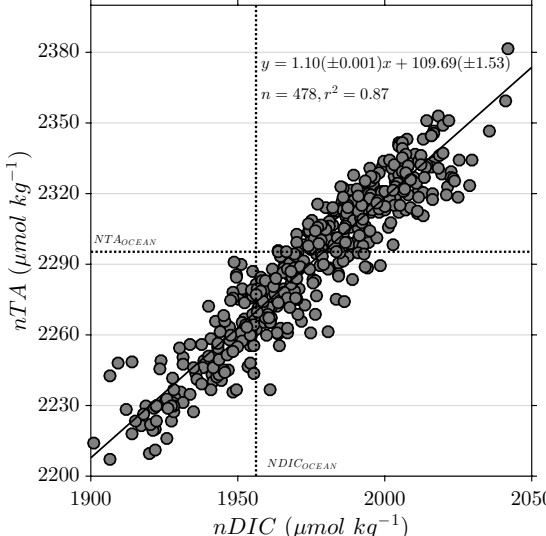

327

**Figure 6: Relationship between salinity-normalized DIC (nDIC; μmol kg$^{-1}$) and salinity-normalized TA (nTA; μmol kg$^{-1}$) using the *in situ* bottle measurements collected at the MapCO$_2$ buoy from 2009 to 2017. DIC and TA measurements were normalized by the mean oceanic salinity (S= 36.0) to correct for the influence of freshwater addition and removal; and explore the biological processes that also influence TA and DIC.**



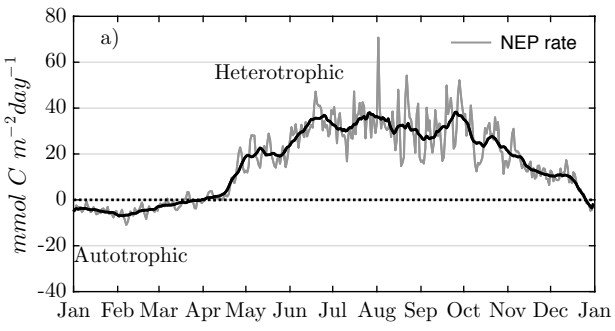

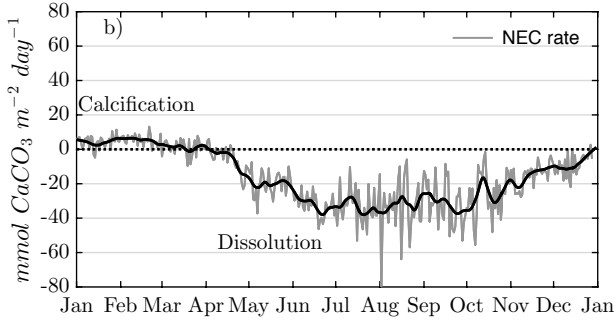

**Figure 7: Annual composites based on daily averages of modeled a) NEP (mmol C m⁻² day⁻¹; gray line) and b) NEC rates (mmol CaCO₃ m⁻² day⁻¹; gray line). Solid black line is a moving average of span 15. NEP > 0 are representative of net heterotrophic and NEP < 0 indicate net autotrophic. NEC > 0 indicates net calcification and NEC < 0 net dissolution.**

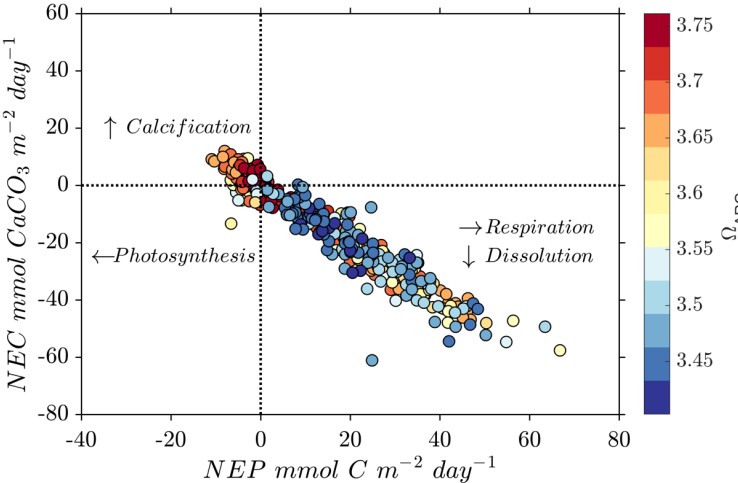

**Figure 8: NEC (mmol CaCO₃ m⁻² day⁻¹) versus NEP rates (mmol C m⁻² day⁻¹) in relationship with Ω_arag indicated with colorbars. The zero line of NEC and NEP are denoted by grey dashed lines.**





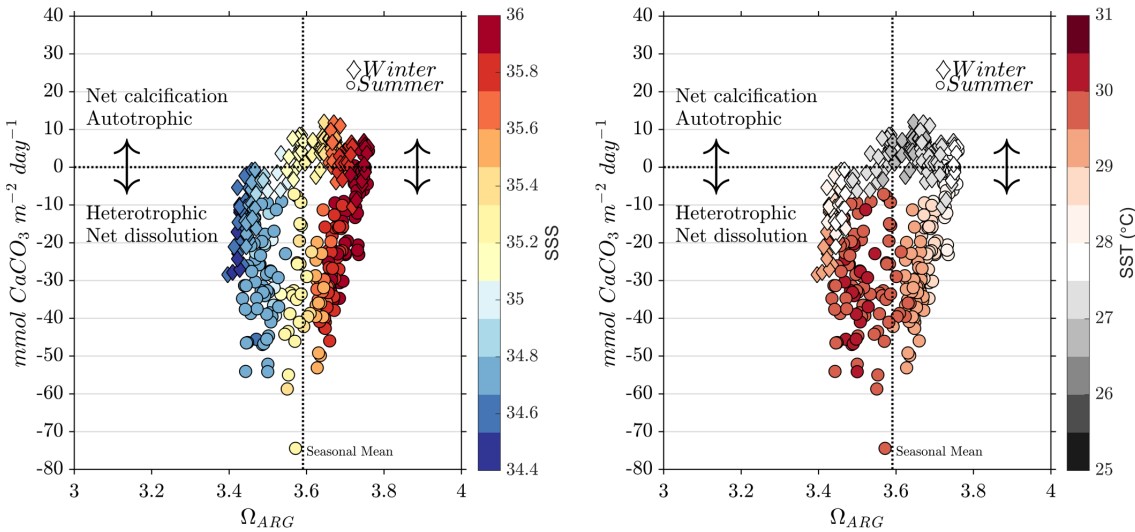

340

**Figure 9: Seasonal hysteresis of NEC (mmol CaCO₃ m⁻² day⁻¹) relative to Ω_arag and colored by (a) SSS and (b) SST (ºC) indicated**
**with colorbars. The diamond symbols represent the winter values and the circle symbols the summer. The zero line of NEC and**
**the seasonal mean of Ω_arag are denoted by grey dashed lines.**

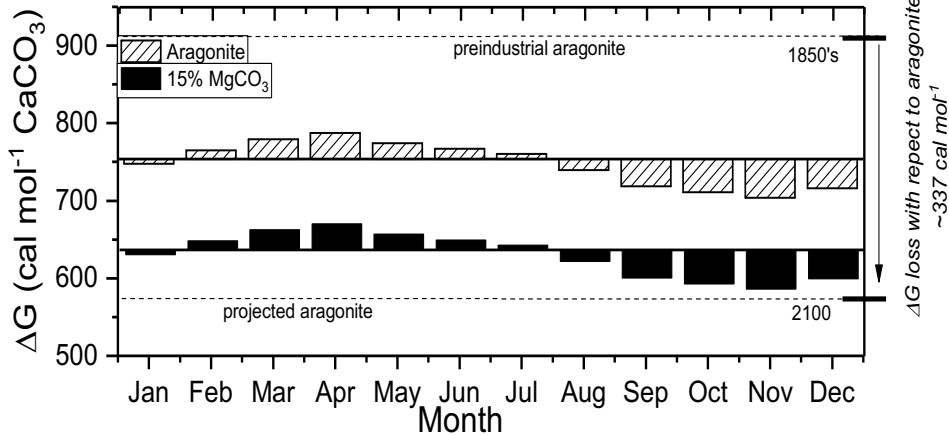

344

**Figure 10: Seasonal dynamics in free energy (cal mol-1 CaCO3) at Enrique reef for Aragonite (dashed) and 15% MgCO3 (solid**
**black). The dashed lines represent the free energy for aragonite in the 1850 and the projection for 2100.**