# Peer review of "S1. Atlantic Ocean Acidification Test-bed (AOAT)"

_Biogeosciences, 2018_

## Referee Comment (RC1) · Anonymous Referee #1 · 30 Nov 2018

The presented study has a beautiful dataset of time series measurements over a heterogeneous nearshore environment in Puerto Rico. The manuscript introduction and discussion are clear and well-written and the data produce visually convincing and yearly integrated rates that are rare for these environments. However, I have a number of methodological and technical concerns about the way the data was analyzed and applied, which make the validation of the presented 1D model and its results difficult.

A major issue that I have with this study is the focus on coral reefs and whether the presented 1-D mass balance is reflective of coral reef processes. The monitoring location is on the ocean side of a fringe reef with the mean current coming from offshore and what appears to be a relatively steep reef slope. Thus, concentration changes are indicative of the upstream water column processes and benthic communities while the

results are extrapolated to the downstream fore reef which occupies a relatively small area. Without data on the footprint of the 1D mass balance, current directions, and flow rates it is challenging to generalize these results to a very heterogeneous area that has shallow fringe reefs, sand flats, seagrass beds, mangroves, and deeper environments all adjacent over ∼0.5km distance, especially considering the authors estimate of concentration contributions from up to 6km away (Ln 498). Daily and seasonal changes in current and wind direction (which are present according to Ln 498-499) could significantly bias results that are largely extrapolated to reef processes. For example, if the currents/winds come from the North, a large portion of the signal will come from the large upstream seagrass bed (Figure 2c).

I am particularly concerned with how appropriate the chemical assumptions are (2:1 TA:DIC calcification and 1:1 DIC:O2 metabolic ratios) when the footprint is situated over non-reef environments? How applicable are these to water column processes (see later comments)?

Another major concern is that the main tracers in the study (pCO2 and O2) are treated differently. For example, in Ln 357 and Ln 438, pCO2 and O2 are treated differently in that no advective or "HOR MIX" term is applied for O2. How can horizontal mixing be ignored for O2 and be treated differently than CO2? Certainly this requires some discussion, explanation, and validation. Also, see other comments on bubble corrections being applied to only O2.

Further, much of the results are obtained from an oxygen sensor that has limited marine application and has not been validated sufficiently (Ln 293-298). This section is a bit misleading, as it presents the "best fit" for calibration of the sensors (n=40) whereas a dataset exists that is 500 times larger and is presented in the supplement (n = 21456). It is not clear which calibration is used (Figure S1 or S2). It seems an overall calibration should be reported, not the "best" one. Further, an R2 of 0.78 is poor for a calibration curve with so many data points. I also wonder if a salinity/temperature correction would improve these results? At times the variability around the fit curve is 145-195 umol kg-

1 (Figure S2) while the total variability is from 145-215 umol kg-1. Unfortunately, it may not be possible to determine which sensor or data are correct given the variability and issues brought up with the sensors used (Ln 272-298). Further, it is not clear why RMSE are reported in Figure S1 and S2 calibrations and p values are reported for calibrations in S3. Overall, this greatly reduces my confidence in the presented O2 values and fluxes and much more work is needed here to validate these numbers, especially since the authors are using a non-standard O2 sensor that is not designed for seawater measurements. Finally, the referenced study that previously used this sensor (Xue et al. 2016) only state that it is used in conjunction with Chl A data and "can be used to roughly reflect biological activity in combination with DO% data", which is not a strong endorsement for the use of this sensor.

Ln 382-397. This section on physical transport leans fairly heavily on the assumptions of Xue et al. 2016 and much more details (at least in the supplement) are needed such as the DIC-S slope vs. discharge relationships from Xue et al., 2016. Seeing that the authors note that there is limited freshwater input (Ln 503) I wonder how the assumptions from Xue et al. fit here, which assume a freshwater end-member? The salinity at the buoy averages 35.3 (Table 2); I wonder what the offshore salinity was and if this method can even be applied / is appropriate for in this environment? It would seem that this physical transport term is negligible from Figure 5; is this due to the fact that there are negligible salinity differences? Therefore, is this even a good indicator of horizontal transport at this site?

Additionally, (Ln 382-386) when were the cruise data from that parameterized the off-shore end-member and how were they distributed through the year? How were the concentrations interpolated through the year?

Ln 444 The O2:DIC ratio is stated as 1.1, but I cannot find this data. Applying this ratio from simple time-series measurements is not straightforward as calcification influences the DIC. More discussion / data is need to validate this, especially considering that any variability in this ratio will affect the NEC rates as defined in Eq. 13. Much more

discussion and validation seems needed to apply this stoichiometry, especially in this heterogeneous environment where some communities can have DIC:O2 ratios that vary significantly from 1 (seagrass, sands, water column, etc.).

Ln 621-630. It is finally addressed here that these measurements are not benthic fluxes (i.e. coral reef fluxes) but are integrated across the water column and benthos (over a large, undefined heterogeneous area). Yet, most of the generalization in the paper are for coral reefs. This discussion / caveat should come much earlier and be further explored. For example, assuming the 6km foot print, how does water depth vary over this area (e.g. the relative importance of water vs. benthic), how does the benthic community vary over this area (e.g. sand, grass, reef?), what is the magnitude of the advective term through time, are there any seasonal changes in tides or wind that may cause a different model footprint?

Ln 621-630. The presented 1-D mass balance is very different from incubations (very small area), mesocosms (ex situ), gradient flux (benthic only), Eulerian (defined control volume), Lagrangian (follows a water parcel) – I am not sure the point trying to be made here is but this over simplifies many important differences between these methods. I suggest to delete this whole section.

Ln 646-659. Related to a comment above, this section suggests that water column DOM and POM may be driving the observed changes. I would interpret this as a potential significant influence of water column processes. This should be explored more and discussed more explicitly as differences between benthic and water column processes. See earlier comments.

Overall, many of these issues have been identified and addressed through contrasting "open-water" or 1D-type mass balances to the more advanced techniques cited in this study (Ln 174-177). These techniques (incubations, gradient flux, Eulerian, Lagrangian, etc.) were developed due to the limitations of inferring benthic or water column processes from a 1D balance of time rate of change measurements, especially

in coastal heterogeneous environments. Unfortunately, with simple time-series measurements and 1D mass balances these methodological biases and caveats remain, and must be addressed and reflected in the presented results and conclusions.

Addressing these concerns will likely lead to a re-analysis of stoichiometry, concentrations and fluxes in the manuscript, a significant re-focusing of the intro and discussion away from reef-specific processes, and a detailed introduction and discussion on the limitation of the use of a 1D model in this environment.

Detailed comments:

Ln 153 – The TA to DIC ratio is reversed here; it should be 2(TA):1(DIC). Considering this, I find that it would be clearer to write Eq. 2 as: $Ca2+ + 2HCO3- > CaCO3 + CO2 + H20$ to provide the complete stoichiometry and highlight the origin of this 2:1 ratio that is a central tenet to the presented chemistry.

Ln 154 – "DIC:TA ratio" is inconsistent with previous sentence (but consistent with the ratio). I find that the TA:DIC ratio is most appropriate considering the suggested re-write of Eq. 2.

Ln 294. The presented r2 is different than shown in Fig S1. The units are also different (umol kg-1 vs mmol m3). It is not clear what calibration is used for what periods. Figure S2 shows a reduced correlation coefficient, and significant variability that greatly reduces the confidence of the O2 calibration and measurements.

Ln 346-348. This sentence is very awkward.

Ln 376-378. It would be interesting to see the data from the temperature loggers. I find it unlikely that stratification never occurred, but some simple metrics between the top and bottom sensors could easily demonstrate this and possibly it's insignificance (e.g. <1% of time).

Ln 380. Mixing due to a 0.25m tide in a 3m deep water column may not be insignificant (8%) especially when compared with the previous sentence (<0.1%). Also, I would

assume that some of the horizontal advection is due to tides. Is the horizontal advection assumed to be only due to wind-driven currents?

Ln 403 (Eq9) and ln 410. The use of "mixed layer water depth" seems odd here considering the water is only 3m deep. It would appear that Eq. 9 needs to be re-parameterized for shallow water environments as it was previously stated that there was no stratification.

Ln 440 –O2 units in mmol m-3 while DIC units are umol kg-1, Please be consistent.

Ln 441-442 "and corrected by the bubble flux injection and the bubble flux exchange (Manning and Nicholson, 2016)" This reference is a bit obscure because it is just a link to MATLAB code without further method explanation. Was this correction done for CO2 in section 2.6.3? I find it odd that CO2 and O2 are note treated the same in terms of air-sea flux. These should be condensed into a single section on air-sea exchange and treated the same.

Ln 490-506. I am confused by this section. In the case of a 1D balance the footprint is simply a function of water residence time, which the authors calculate from the flow velocity. The part that is confusing is starting with a discussion on spatial variability with a unit of change per time (and not space!). Spatial variability should be examined at the same time, or time-corrected for diel variability, and reported as +/- pCO2 over the 2km transects. This section should be re-written or deleted.

Ln 491-492. What was the variability in the other parameters (SSS, SST, O2)? Was it consistent with the CO2 variability? How did the O2 measurements conducted at this time compare to the MaxTec O2 gas sensor?

Ln 491-496. Is this the only time period when currents were measured? Did the measurements capture a full tidal cycle? How indicative of yearly flow rates is this? What about seasonal changes in wind direction?

How and from what data was the mean current direction in Figure 2 calculated? Did

flow always come from offshore? Were there changes in the current direction over a tidal cycle?

Ln 494 "averaged column velocity profiles" it is not clear what this is.

Ln 501-502. "Enrique's physicochemical" to Enrique Reefs physical-chemical

Ln 509 "On an annual scale, the Enrique reef experiences a seasonal SST daily average variations" These 3 conflicting time-scale adjectives are very confusing.

Ln 521 "Enrique forereef, like many other reefs, is a persistent source of $CO_2$ to the atmosphere ($2.04 \pm 2.13$ mmol $CO_2$ m-2 day-1)" Based simply on the presented SD, I would argue it is balanced and/or not different from zero. Since many of these net rates are around zero with high SD I would suggest some additional statistics to support these conclusions.

Ln 527-528 "The injection of bubbles represents <2 % of the total $O_2$ flux variation at the site." I would move this up to the methods and say it is not important and remove it from analysis. Since $O_2$ is much less soluble than $CO_2$, the same conclusion can be assumed for $CO_2$. See earlier comments.

Ln 559 nTA and nDIC are not defined.

Ln 577 see earlier comments on mixed layer

Ln 589 "of the method is that is based on" add it

Ln 690-691. Applying rates determined over a 6km footprint to such a small (and benthic) area is an invalid comparison. It is likely that the forereef rates in this small area (0.07km2) are very different. See earlier comments.

Ln 695-704. Water column processes are ignored in this discussion on benthic reef processes and this section does not address the limits of the presented dataset.

Ln 838-839. "for providing the Be-7 sampling and residence time analyses." Where is

this data? It could help explain some of the issues with physical transport and ecosystem attribution (see earlier comments).

Figure 2 – please add scale bars to (and letters) to figures. Lajas is not shown.

Fig S1 and S2 – Please provide p values as is done in S3.

Please add plots of the measured/modeled parameters (SST, SSS, TA, DIC, pH) vs the in situ bottle samples so the reader can evaluate how accurate the modeled and bottle samples are. This could go in the supplement or at least report correlations and p values.

---

## Referee Comment (RC2) · Anonymous Referee #2 · 18 Dec 2018

This paper presents estimates of NEC and NEP on a reef in Puerto-Rico based on continuous monitoring of pCO2 and O2, and discrete bottle sampling for TA and DIC. The authors used a large dataset and applied a simple 1-D model to estimate the metabolic rates of the reefs. The main result is that the reef is currently dissolving at a rate faster than what has been estimated before using other methods. This result is highly interesting and also shows that other methods such as the reef budget of Perry et al. should be used with caution. The methods used seem to provide reliable results even if large errors in the estimates of TA are problematic. This problem will need to be overcome, likely by increasing the frequency of sampling in further research. The paper is well written and the data presented are of interest for biologist and biogeochemist. However, I regret that the paper is that long. I do understand that some details were

needed but I believe that a shorter version of this paper would attract a broader readership. For example the discussion is rather long (∼10 pages) with some repetitions. The introduction could also be shortened by maybe not providing trivial information on carbonate chemistry.

I have listed below some specific comments:

-Introduction: The two first pages could be shorten

- L133-134: It would be good to add 1-2 sentences on the poor coral cover/health of Caribbean reefs here.

- 160-161: The link NEP NEC is not clear here, why "relative to NEP"?

- Methods: The method section is a bit confusing as some parts read more like discussion/ result material (for example L 249-253). I recommend reformatting this section to make it a bit easier to read by removing all the materials that is not methods.

- L317-319: Please provide more details on the methods used to determine TA and pH (accuracy, etc).

-L336-338: The errors with this method are very large and could potentially bias all further results. Looking at Fig S3 it looks like for a given salinity it is possible to obtain a TA range of up to 200 umol kg-1... It would be good to discuss this potential pitfall in the discussion.

- L480-489: again an example of a section that has nothing to do in the methods.

-L517: Did you measure any seasonal changes in phosphate and silicate?

-L599: What about the changes in coral cover between studies?

-L611: Are they any other major calcifying organisms at this site? What about CCA, Halimeda, or forams that can contribute massively to NEC?

-L661-663: What about the role of temperature. Could these results also demonstrate

that 1) Corals calcify more slowly when temp > 27, and 2) that bacterial activity is enhanced by increasing temperature which favour the dissolution of sediment, etc. in interaction with increasing DOM. It is also interesting to see that there is maybe no relationship NEP

–nutrient, could that demonstrate that one critical nutrient is missing in the system (e..g Iron)?

-L683-684: This decoupling between omega and NEC is very interesting. The role of SST on the biological activity is probably very important here (see my previous comment).

-L712-715: This is a critical point. Is there any reason to believe that Enrique reef is a "special case" or is it likely to observe the same discrepancy on other reefs?

-L728: Where does that come from? This claim needs a reference because the link between net heterotrophy and algae dissolution is not clear.

-Section 4.5: I am not sure about the utility of this section. The manuscript is already rather long and this section reads like another story.

---

## Author Comment (AC1) · 19 Feb 2019

- 9 Anonymous Referee #1
- 10 Received and published: 18 December 2018
- 11

12 The presented study has a beautiful dataset of time series measurements over a heterogeneous

13 nearshore environment in Puerto Rico. The manuscript introduction and discussion are clear and

14 well-written and the data produce visually convincing and yearly integrated rates that are rare for

15 these environments. However, I have a number of methodological and technical concerns about

- the way the data was analyzed and applied, which make the validation of the presented 1D model
- 17 and its results difficult.
- 18

19 A major issue that I have with this study is the focus on coral reefs and whether the presented 1-20 D mass balance is reflective of coral reef processes. The monitoring location is on the ocean side 21 of a fringe reef with the mean current coming from offshore and what appears to be a relatively steep reef slope. Thus, concentration changes are indicative of the upstream water column 22 processes and benthic communities while the results are extrapolated to the downstream fore reef 23 which occupies a relatively small area. Without data on the footprint of the 1D mass balance, 24 25 current directions, and flow rates it is challenging to generalize these results to a very 26 heterogeneous area that has shallow fringe reefs, sand flats, seagrass beds, mangroves, and 27 deeper environments all adjacent over ~0.5km distance, especially considering the authors 28 estimate of concentration contributions from up to 6km away (Ln 498). Daily and seasonal 29 changes in current and wind direction (which are present according to Ln 498-499) could significantly bias results that are largely extrapolated to reef processes. For example, if the 30 currents/winds come from the North, a large portion of the signal will come from the large 31 32 upstream seagrass bed (Figure 2c).

33 34

35 We appreciate your comments. We agree the observations are not solely reflecting Enrique coral 36 reef metabolism and that attributing the estimated processes to "Cayo Enrique mid shelf coral 37 reef" could be misleading. Our study does not attempt to evaluate the role of any particular 38 benthic community on NEC or NEP variability. The primary objective of this study is to 39 characterize the temporal carbonate chemistry changes observed by the MapCO2 buoy and to 40 discern the predominant biogeochemical and physical processes that drive said variability. One 41 caveat of this study that should have been stated more clearly, is that it does not provide the relative contributions of different benthic community types to NEC or NEP. We agree that this 42 43 needs to be clarified in the introduction to clearly state that the waters the buoy observes are 44 affected by coastal physical and biological processes associated to the shelf ecosystems of La 45 Parguera, that is indeed comprised of mangrove forests, seagrass beds, unconsolidated 46 sediments, coral reefs, hard bottom carbonate substrates, and phytoplankton communities. This 47 will be clearly stated in the introduction of the revised manuscript. However, although the

48 primary objective of the Atlantic Ocean Acidification test-bed is to monitor near-reef carbonate

- 49 chemistry and explicitly account for the effects of OA and determine its impact on coral reefs,
- 50 this study offers new possibilities to gain meaningful insight into the biogeochemical processes
- 51 occurring in coastal marine environments and which can significantly modulate said impact.
- 52 Furthermore, we believe users of the existing observational OA assets data will benefit from
- application of methods presented to develop further understanding of ecosystem metabolic
- 54 processes.
- Efforts to better understand the hydrodynamics in the area and the extent of the buoy's footprintshould be an essential component of the buoy's observations. A better understanding of how the
- 57 hydrodynamics (e.g., currents/winds) change the footprint and how different functional groups
- 58 affect the disequilibrium between coastal and open ocean waters are essential questions raised
- 59 from this study. However, further observations on the hydrodynamics, residence times, organic
- 60 carbon sources, benthic and fish communities are needed to fully answer these questions.
- 61 We agree that this issue requires attention. For this revision we will provide a conservative
- 62 footprint estimation and re-write the section 2.11 that explains the area over which our
- 63 measurements are influenced (see below). We will also provide a table that shows % cover of the
- 64 different benthic communities and the scaled NEC and NEP presumably attributable to each
- 65 benthic type.
- 66
- 67 Preliminary results on the footprint using Acoustic Doppler Current Profiler (ADCP) located
- about 0.20 km south of the buoy shows that the line of the extent of the footprint is
- approximately 2.63 km from the North East and 1.43 km from the South East (Fig.1). The two
- 70 major current components are 3.38 cm/s, 290° and 6.13 cm/s, 140°. We scaled up to the tidal
- 71 period of 12 hr according to the methodology described by Courtney et al. (2016). This method
- 72 assumes the flow is tidally driven. The primary author did a spectral analysis to check the period
- of the winds and the currents, and the dominant period is coherent with the tides, which gives us
- a good measure of the timescales over which the footprint would be defined. The orthogonal (orside) components of these currents it is challenging to determine due to the weak eastward flow
- 75 side) components of these currents it is channeliging to determine due to the weak eastward now 76 and the "channel" (between two reefs) nature of the location where the ADCP was positioned
- and the challer (between two reels) nature of the location where the ADCI was positioned and where the buoy is located (Fig. 1). Additional evidence of this weak eastward flow from
- 78 hydrodynamic observations in La Parguera (date from 1997) showed that occasionally, the
- reastward tidal component could not overcome the mean westward flow resulting in
- 80 acceleration/deceleration of the westward flow rather than causing east-west reversals
- 81 (unpublished observations). We note that the bathymetric features relative to our buoy asset
- 82 does not support the use of the Principal Component analysis (the method used in Courtney et al.
- 83 (2016)) to describe the footprint.
- 84
- 85 In the revised manuscript we will provide an estimate of the extent of the footprint using
- 86 available ADCP current velocity measurements adjacent to the buoy (March 2017, November
- 87 2016, and from February June 2009). The benthic data will be analyzed to show the % cover of
- the different benthic communities. This data is available through the NOAA Biogeography
- 89 Branch (Bauer et al. 2012).

90 91

92 Figure 1: Mean surface currents at Enrique during November 2016 and the corresponding lines
93 of the extent of the footprint. The white dot indicates the buoy's location. The width of the lobe
94 is the unknown (yellow color).

I am particularly concerned with how appropriate the chemical assumptions are (2:1 TA: DIC

97 calcification and 1:1 DIC: O2 metabolic ratios) when the footprint is situated over non-reef98 environments? How applicable are these to water column processes (see later comments)?

98 99

We will add a sentence on the discussion about this caveat and the assumptions made in thisstudy. It might be important to note that recent work has begun to demonstrate that Redfield may

102 not hold, and in fact may vary, within coral reef ecosystems (e.g., Rosset et al., 2017). For

103 Enrique forereef and Enrique seagrass relative to the

104 offshore station, we observed a mean  $\Delta TA/\Delta TCO_2$  ratio of

105 0.7 and 0.4, respectively (Fig.2). The depletion of TA was

- 106 calculated as the difference between reef and offshore TA
- 107 values. It is important to note that this offshore station is 10
- 108 km away from Enrique. In coral reef environments, where
- 109 calcification is dominant (but not the only) process
- 110 affecting seawater chemistry,  $\Delta TA/\Delta TCO2$  is near 0.5
- (e.g., Cyronak et al., 2018). Our observations suggest thatthe TA and DIC behavior in the forereef of Enrique is
- 112 the TA and DIC behavior in the forefeet of Enrique 1

indicative of a system where calcium carbonate

calcification/dissolution processes dominate. Figure 2shows that major metabolic and biogeochemical processes

are shown with the calcification path represented on the

- $A_{\rm T}$ -DIC diagram as a slope of 2. While calcification is an
- 118 important process throughout much of the year, respiration

119 particularly in the late spring appears be an additional

120 source of DIC to the system.

- 121
- 122
- 123
- 124
- 125

Figure 2: Changes in TA and DIC concentration (normalized to S=35) between Enrique reef, seagrasses relative to offshore waters.

- The slope of the corrected O2 measurements against DIC at the buoy site shows a slope 1.1 126
- (Fig.3) with a weak linear correlation coefficient of  $r^2 = 0.35$ , but significant (p-value<0.0001, n 127
- = 28340). Variation in this molar stoichiometry (i.e. the P/O ratio) can arise in certain 128
- environments if organic carbon production is coupled to significant uptake of  $NO_3^-$  or  $NH_4^+$ , but 129
- 130 this ratio is typical of many other reef ecosystems (e.g., Crossland et al., 1991).
- 131
- 132 Odum et al., (1959) measured the metabolism in Enrique reef using upstream and downstream
- methods and found that the photosynthesis to respiration ratio was 1.15. The concentrations of 133
- dissolved inorganic nutrients based on the concentration of nitrate and phosphate (< 0.03 uM) in 134 the area suggest that the impact of skewed stoichiometry ratios is less pronounced. Potential

- 135
- 136 deviations of these stoichiometries can
- change the NEC and NEP absolute rates, 137 but not the major seasonal dynamics. We 138
- 139 agree these ratios can vary in daily time
- scales, depending on how long a single 140
- 141 community within the footprint affect the
- 142 buoy measurements. However, this doesn't
- 143 change our model's results or conclusions
- but would change the numbers slightly. 144
- 145
- 146 The stoichiometry assumptions may
- 147 produce errors that are negligible because of
- 148 the large natural variability of CO2 system
- parameters. However, we will add the NEC 149
- and NEP changes associated to these 150
- 151 stoichiometry assumptions on the
- 152 discussion of the revised manuscript.

---

## Author Comment (AC2) · 19 Feb 2019

This paper presents estimates of NEC and NEP on a reef in Puerto-Rico based on continuous monitoring of pCO2 and O2, and discrete bottle sampling for TA and DIC. The authors used a large dataset and applied a simple 1-D model to estimate the metabolic rates of the reefs. The main result is that the reef is currently dissolving at a rate faster than what has been estimated before using other methods. This result is highly interesting and also shows that other methods such as the reef budget of Perry et al. should be used with caution. The methods used seem to provide reliable results even if large errors in the estimates of TA are problematic. This problem will need to be overcome, likely by increasing the frequency of sampling in further research. The paper is well written and the data presented are of interest for biologist and biogeochemist. However, I regret that the paper is that long. I do understand that some details were needed but I believe that a shorter version of this paper would attract a broader readership. For example the discussion is rather long (_10 pages) with some repetitions. The introduction could also be shortened by maybe not providing trivial information on carbonate chemistry.

We agree with the reviewer that the paper is long, and have modified the introduction, methods, and discussion accordingly. We also removed the section on "free energy". The specific comments are answered below.

I have listed below some specific comments:

-Introduction: The two first pages could be shorten

We appreciate you bringing this to our awareness, and we have shortened the introduction accordingly. L61-93 will be summarized.

- L133-134: It would be good to add 1-2 sentences on the poor coral cover/health of Caribbean reefs here.

Agree. The text will be added to L133-134 accordingly to explain the declines in hard coral cover and increase in the abundance of the macroalgae over the last 30 years in the region (Gardner et al. 2003). These ecosystem changes are related to the coral mortality from diseases, depletion of herbivorous fishes and the black sea urchin (*Diadema antillarum*), bleaching events and the interaction of multiple anthropogenic stressors such as fishing and sedimentation.

- 160-161: The link NEP NEC is not clear here, why "relative to NEP"?

We have revised the text and modify the sentence by deleting "relative to NEP" and added relative to dissolution processes to clear up the confusion.

- Methods: The method section is a bit confusing as some parts read more like discussion/result material (for example L 249-253). I recommend reformatting this section to make it a bit easier to read by removing all the materials that is not methods.

We will modify the text accordingly so that the methods are easier to follow. Ln 249 -253 were moved to the discussion section.

- L317-319: Please provide more details on the methods used to determine TA and pH (accuracy, etc).

We will add the pH and TA analysis accuracy to Ln 318-319 and modify the text accordingly. The details of the sample collection, analysis, and accuracy are in the supplemental material, section 2 (S2). We chose the supplemental material for these details to shorten the manuscript.

-L336-338: The errors with this method are very large and could potentially bias all further results. Looking at Fig S3 it looks like for a given salinity it is possible to obtain a TA range of up to 200 umol kg-1. . . It would be good to discuss this potential pitfall in the discussion.

We will bolster the discussion of uncertainties. Errors on the TA daily variability at the site arise from the lack of a salinity-dependence on coastal processes such as precipitation and dissolution of carbonates at the site. The use of seasonal TA reduces the potential biases caused by biological processes. Figure 1 the bottle data (TA and DIC) grouped in a composite seasonal cycle and compared with the TA and DIC modeled. Both the modeled and bottle data show similar seasonal dynamics. The mean difference between the average of both vectors is a reasonable 11 and 9 umol kg$^{-1}$ for TA and DIC, respectively. However, there are limitations due to the interannual changes on TA, which is shown in the standard deviation of the bottle measurements.

The uncertainty of each parameter on the linear relationship between TA and SSS were added as sources of error to the Monte Carlo simulations, so that all random errors are accounted for. This method is preferred over the "propagation of error" because it considers all each individual error that contributes to the total uncertainty. We think the TA Monte Carlo simulations represent reasonable estimates of the TA uncertainty and the potential bias in NEC. To check this, I compared the potential changes on NEC by using the TA and DIC bottle data in the 1-D model. Both approaches (modeled and bottle data) yield considerable results on NEC with an average difference of 1 mmol m$^2$ day$^{-1}$ (see Table 1). The NEC difference between the modeled and bottle data is considered not statistically different at p-value<0.05. Finally, we point out that the TA is only one term in the estimate of NEC and that the Monte Carlo analyses of the suite of terms show our estimates to be reasonable.

We will provide two additional sentences to the L480-489 about the effect of bottle samples on the NEC rates. This paragraph will be moved to the discussion section.

[Figure]

Figure 1: Modeled (black) and bottle (red) Total Alkalinity and Dissolved Inorganic Carbon.

Table 1: NEC calculated using the model and bottle DIC and TA measurements. Average ± std.

| NEC | | | |
|---|---|---|---|
| units | Bottle TA and DIC | Modeled TA and DIC | Difference |
| mmol m$^2$ day$^{-1}$ | -17.26 (87.25) | -16.21(16.81) | 1.06 |

95% confidence interval of this difference: -8.07 to 10.17 mmol m2 day-1, SE of difference = 4.65 mmol m$^2$ day$^{-1}$. The mean is not statistically different at p-value < 0.05.

- L480-489: again an example of a section that has nothing to do in the methods.

Agreed. We have revised the text, and the L480-489 about the uncertainties on TA will be moved to the discussion section.

-L517: Did you measure any seasonal changes in phosphate and silicate?

We did not measure seasonal changes in phosphate and silicate at the site. Phosphate and Silicate concentrations were collected at the buoy site and measured six times over the period from 2009 to 2011 in January, February, March, May, November, and December. However, there is unpublished data from Caribbean Time Series (CaTS), about 51 km south of the station, that was occupied monthly from May 1996 to June 2007. There is no seasonal change in nitrate, and the concentrations are < 0.03 uM. The phosphate shows a maximum in February and May with concentrations no greater than 0.03 uM. Silicate concentration displays a seasonal cycle with a peak from August to January of ~2.5 uM and lows of <1.5 uM the rest of the year. The potential contribution of these nutrients to TA, assuming this seasonal cycle and concentration of silicate and phosphate, are insignificant (<0.01%) and therefore neglected.

-L599: What about the changes in coral cover between studies?

Currently, a benthic habitat map has identified most of the submerged habitats for La Parguera (Pittman et al. 2010). However, reports on the seasonal changes on the coral cover at the buoy site are scarce. According to Moyer et al., (2012), who did seasonal benthic habitat characterizations at two reefs (including Enrique) and one seagrass site at La Parguera in 2011, reported that fleshy macroalgae dominated both reef sites, and live coral cover ranged from 8 to 10% in all seasons. Manzello et al. (2017) reported 11 % live coral cover, 17 % macroalgae and turf, 26% soft coral and 34% rubble rock and sand from a single survey performed in August 2015 at Enrique reef.

The highest coral cover in La Parguera is observed southward Enrique along the shelf-edge and westward Enrique. Pittman et al. (2010) analyzed the community compositions from 937 sites and covered an area of 93.7 $km^2$ from 2001-2007 and observed that live coral cover varied significantly among some sampling years, but overall live coral cover decreased over the sampling period (2001-2007). Further benthic monitoring would be needed to determine the seasonal changes in % live coral cover at the buoy site.

-L611: Are they any other major calcifying organisms at this site? What about CCA, Halimeda, or forams that can contribute massively to NEC?

We acknowledge that calcifying organisms other than coral are participating the metabolic processes measured in this study. Manzello et al. (2017) surveyed ~20 $m^2$ at the buoy site and reported 0% cover of secondary (coralline algae and other calcareous encrusters) and sediment producers (calcareous algae such as Halimeda and benthic foraminifera). However, information on the seasonal variation of these functional groups is lacking for this site.

The CCA represented the 1.3 % of the hard-bottom habitats. The relative density of benthic forams is unknown for the studies site. However, results from La Parguera by Pittman et al. (2010) show that calcareous macroalgae (e.g., Halimeda spp., Udotea spp., and Penicillus spp.) were commonly encountered on soft bottom habitats, but their percent (12%) cover was low relative to those of seagrass (28%) and the non-calcareous algae (e.g., Lobophora, Dictyota, and Padina spp). Published data on the Halimeda calcification rates ranged between 0.4 - 1.6 kg $m^{-2}$ $yr^{-1}$ and for CCA about 0.181 kg $m^{-2}$ $yr^{-1}$. The estimated calcification rates for benthic forams is between 0.030-0.230 kg $m^{-2}$ $yr^{-1}$. The presence of these species in La Parguera reef platform will likely influence the annual NEC rate. However, the mean calcification seen in the wintertime at the buoy site (0.17 kg $m^{-2}$ $yr^{-1}$) is small relative to the mean calcification rates of these species.

The first author's master's thesis reports on the TA concentration in reef sediment porewaters down to 20 cm sediment depth on a 10 m transect at the buoy site. The change in TA ranged from 48.8 to 75.7 µmol $kg^{-1}$ $cm^{-1}$ and the annual flux calculated was about 100 umol $m^{-2}$ $yr^{-1}$. This estimate can represent the 18% of the yearly change in carbonate nTA calculated in this study (556 umol $m^{-2}$ $yr^{-1}$). Based on observed changes in porewater TA and estimates of vertical flux of TA in sediment porewaters, preliminary data on carbonate dissolution is estimated as 0.003 mmol $m^{-2}$ $h^{-1}$ in the summer of 2011 at Enrique forereef. The calcium carbonate sediment dissolution rates were small and could represent <1% of the NEC at the site.

-L661-663: What about the role of temperature. Could these results also demonstrate that 1) Corals calcify more slowly when temp > 27, and 2) that bacterial activity is enhanced by increasing temperature which favors the dissolution of sediment, etc. in interaction with increasing DOM. It is also interesting to see that there is maybe no relationship NEP

Agree. Our results show that during the calcification months the temperatures range from 26.6 to 27.9 °C, which could be linked to the optimal temperature in which each species can maintain metabolic rates and coral growth (Marshall and Clode, 2004). The increase in temperature during the summertime can reduce calcification rates and make metabolically expensive to maintain high calcification rates. Also, the high respiration rates can be fueled by the increase in organic matter degradation and associated bacterial activity as a result of high temperatures during the summer. During the summer the high temperatures (>29°C) can cause thermal stress to corals and in some instance bleaching, changing the coral's organic matter fluxes. For example, bleached corals may assimilate POM (to fulfill the energetic requirements in the absence of autotrophy) and release carbon as labile DOM (although other studies show a release of DOC by healthy corals as well).

Even though there is the possibility that low calcification rates and NEP may not be associated, there is evidence that shows moderately high levels of sedimentation, bacterial and bacteriophytoplankton counts in Enrique reef and La Parguera inner-shelf reefs during the summer months (Otero, 2009). Still, we don't know exactly from where this organic carbon comes from in this area. We hypothesized this organic carbon could come from nearby mangroves and autochthonous material. Otero (2009) measured isotopic signatures on $^{13}$C of precipitated particles in different places in La Parguera and found that the content on the organic matter is likely autochthonous from seagrasses or algae. He also measured $^{15}$N and found overall low inputs of anthropogenic nitrogen to the system; however the proportion of N allocated from anthropogenic sources were as high as coastal values in Enrique. These intermediate values found at Enrique could indicate shifts caused by mineralization processes due to in situ microbial processes in the absence of inputs of terrigenous or anthropogenic sources of nutrients. Bacterial abundances (0.4 – 1.7 x $10^6$ mL$^{-1}$) and production (2-35 ugC L$^{-1}$ day$^{-1}$were high for La Parguera compared to the CATS station and other reef areas (e.g., Ferrier-Pages and Gattuso, 1998). These shifts in coral metabolism, organic matter fluxes and remineralization of associated bacterial during the summer months, can potentially affect NEP and NEC.

Text will be added in the discussion (L661-670) to explain the coral optimum temperature range and the potential role of the bacteria activity and temperature on NEP and NEC.

–nutrient, could that demonstrate that one critical nutrient is missing in the system (e.g., Iron)?

We consider the micronutrients such as Iron of secondary importance and on the NEP because the productivity in the Caribbean basin is limited by the availability of the limited macronutrients (e.g., Nitrogen and Phosphorous). See the previous comment on phosphorous and silica nutrients. It has also been shown that atmospheric transport of aerosols from desert regions of Africa supply nutrients, such as iron, to the Caribbean waters (e.g., Prospero et al., 1981; Justiniano-Santos, 2010).

-L683-684: This decoupling between omega and NEC is very interesting. The role of SST on the biological activity is probably significant here (see my previous comment).

Agree. See the previous comment.

-L712-715: This is a critical point. Is there any reason to believe that Enrique reef is a "special case" or is it likely to observe the same discrepancy on other reefs?

Line 802-803 we mentioned: Based on similarities in environmental characteristics, our results suggest that tropical Caribbean reef ecosystems and adjacent regions are exhibiting periods of net dissolution". La Parguera shelf is a calcium carbonate platform with emergent fringing reefs, bank-barrier reefs and submerged patch reefs similar to those observed at other areas in the Caribbean such as Yucatan, Jamaica, and Belize (e.g., Morelock et al., 2001). Moreover, the Caribbean coastal ecosystems are susceptible to similar environmental impacts, in part because of their oligotrophic conditions (e.g., Lapointe 1997), and suffer from similar natural and anthropogenic disturbances (e.g., Gardner et al., 2003; Rivera-Monroy et al., 2004, Alvarez-Filip et al., 2009; Eakin et al., 2010; Chollett et al., 2012). Unfortunately, there is not an optimal study or method with which to compare because most of the available data in the Caribbean does not provide a seasonal overview of the carbon to inorganic carbon balance (NEC and NEP).

Figure 2 was amended from Courtney et al. (2016) to offer a better perspective on where our results lie. The figure shows the summation of calcification and mechanical erosion and does not incorporate dissolution (except Courtney's single NEC chemistry point). Our measurements are located on the graph (red dot, -0.5 kg $CaCO_3$ $m^{-2}$ $yr^{-1}$ or 0.5 mmol $m^{-2}$ $day^{-1}$) show that our estimates are not that far from other methods (bottle and census) used to estimate NEC in reefs with ~10% live coral cover.  For comparing purposes, we assumed the 10% live coral cover present at the buoy site, even though we are somewhat unsure of the spatial extent of our measurements (and hence the % live cover within the active area). Statistically speaking, many of the central tendency measures are not that different than ours because of the high uncertainty. Below are the data for the studies that reported uncertainty measures. For example, if the uncertainty reported is the standard deviation, then there's a ~17% chance that dissolution in kg $CaCO_3$ $m^{-2}$ $yr^{-1}$ will be greater (more negative) than the difference indicated below.  At the uncertainties reported, this study may not be particularly anomalous regarding dissolution. This speaks to the difficulties of such measurements.

[Figure]

Figure 2: Summary plot of Caribbean Reef percent hard coral cover vs. Net Ecosystem Calcification (NEC) adapted from Perry et al. (2013) and modified from Courtney et al., 2016. This figure shows NEC measurements from the census and bottle methods. The red dot is the result of this study. We can add this figure to the supplemental material.

-L728: Where does that come from? This claim needs a reference because the link between net heterotrophy and algae dissolution is not clear.

We have modified the sentence and add a reference to clarify the link between the heterotrophy and dissolution of carbonate sediments. We want to point out that during periods of net heterotrophy, $CO_2$ production can enhance $CaCO_3$ sediment dissolution of highly soluble minerals present in cement, sediments, and organisms such as crustose coralline algae. Reductions in NEC not only depend on the changes of the benthic communities they also can be driven by an increase in dissolution rates that have been enhanced by the metabolic production of $CO_2$ during the respiration associated with inputs of organic matter.

-Section 4.5: I am not sure about the utility of this section. The manuscript is already rather long, and this section reads like another story.

We agreed. This section will be considered for another article. We have deleted this section (L 739-777) and Figure 10 from the discussion in the revised paper.

**References:**

Alvarez-Filip, L., Dulvy, N. K., Gill, J. A., Côté, I. M., & Watkinson, A. R. (2009). Flattening of Caribbean coral reefs: region-wide declines in architectural complexity. Proceedings of the Royal Society of London B: Biological Sciences, 276(1669), 3019-3025.

Chollett, I., Müller-Karger, F. E., Heron, S. F., Skirving, W. and Mumby, P. J.: Seasonal and spatial heterogeneity of recent sea surface temperature trends in the Caribbean Sea and southeast Gulf of Mexico, Mar. Pollut. Bull., 64(5), 956–965, doi:10.1016/j.marpolbul.2012.02.016, 2012.

Courtney, T. A., Andersson, A. J., Bates, N. R., Collins, A., Cyronak, T., De Putron, S. J., ... & Musielewicz, S. (2016). Comparing chemistry and census-based estimates of net ecosystem calcification on a rim reef in Bermuda. Frontiers in Marine Science, 3, 181.

Eakin, C. M., Morgan, J. A., Heron, S. F., et al.: Caribbean corals in crisis: Record thermal stress, bleaching, and mortality in 2005, PLoS One, 5(11), doi:10.1371/journal.pone.0013969, 2010.

Ferrier-Pagès, C., Gattuso, J. P., Cauwet, G., Jaubert, J., & Allemand, D. (1998). Release of dissolved organic carbon and nitrogen by the zooxanthellate coral Galaxea fascicularis. Marine Ecology Progress Series, 172, 265-274.

Gardner, T. A., Cote, I. M., Gill, J. A., Grant, A. and Watkinson, A. R.: Long-Term Region-Wide Declines in Caribbean Corals, Science (80-. )., 301(5635), 958–960, doi:10.1038/020493a0, 2003.

Justiniano-Santos, A. 2010. Influence of Saharan Aerosols on Phytoplankton Biomass in the Tropical North Atlantic Ocean. Ph.D Thesis, University of Puerto Rico, Mayaguez, Marine Sciences.

Lapointe, B. E. (1997). Nutrient thresholds for bottom-up control of macroalgal blooms on coral reefs in Jamaica and southeast Florida. Limnology and Oceanography, 42(5part2), 1119-1131.

Manzello, D., Enochs, I., Valentino, L., Kolodziej, G., Carlton, R., Jones, P.: National Oceanic and Atmospheric Administration; Cooperative Institute for Marine and Atmospheric Studies. National Coral Reef Monitoring Program: Carbonate Budget data of Cayo Enrique near La Parguera, Puerto Rico from 2015-08-03 to 2015-08-07 (NCEI Accession 0157740). Version 1.1. NOAA National Centers for Environmental Information. Dataset, 2017.

Marshall, A. T., & Clode, P. (2004). Calcification rate and the effect of temperature in a zooxanthellate and an azooxanthellate scleractinian reef coral. Coral reefs, 23(2), 218-224.

Morelock, J., Ramírez, W. R., Bruckner, A. W., & Carlo, M. (2001). Status of coral reefs, southwest Puerto Rico. Caribean Journal of Science, special publication, 4, 57.

Moyer, R. P., Viehman, T. S., Piniak, G. A. and Gledhill, D. K.: Linking seasonal changes in benthic community structure to seawater chemistry, (July), 9–13, 2012.

Otero, E. (2009). Spatial and temporal patterns of water quality indicators in reef systems of southwestern Puerto Rico. Caribbean Journal of Science, 45(2–3), 168-180.

Perry, C. T., Murphy, G. N., Kench, P. S., Smithers, S. G., Edinger, E. N., Steneck, R. S., & Mumby, P. J. (2013). Caribbean-wide decline in carbonate production threatens coral reef growth. Nature communications, 4, 1402.

Pittman, S. J., Hile, S. D., Jeffrey, C. F., Clark, R., Woody, K., Herlach, B. D., ... & Appeldoorn, R. (2010). Coral reef ecosystems of Reserva Natural La Parguera (Puerto Rico): Spatial and temporal patterns in fish and benthic communities (2001-2007). NOAA Technical Memorandum NOS NCCOS, 107, 202.

Prospero, J.M., Glaccum, R.A. and Nees, R.T. 1981. Atmospheric transport of soil dust from Africa to South America. Nature, 289, 570–572.

Rivera-Monroy V., Twilley R., Weil, E.: A conceptual framework to develop long-term ecological research and management objectives in the wider Caribbean region, Bioscience, 54(9), 843–856, doi:10.1641/0006-3568(2004)054[0843:ACFTDL]2.0.CO;2, 2004.